# TextAtlas5M: A Large-scale Dataset for Dense Text Image Generation

**Dongxing Mao** [* 1]   **Alex Jinpneg Wang** [* 1]   **Weiming Han** [2]   **Jiawei Zhang** [2]   **Zhuobai Dong** [1]   **Linjie Li** [3]   **Yiqi Lin** [4]
**Zhengyuan Yang** [3]   **Fuwei Zhang** [2]   **Lijuan Wang** [3]   **Min Li** [1]

## Abstract

Text-conditioned image generation has made rapid progress, yet rendering images with long-form text remains challenging due to the limitations of existing datasets, which predominantly focus on short and simple text. We introduce TextAtlas5M, a large-scale dataset designed to evaluate *long-text rendering*, where "long text" encompasses not only textual length but also layout complexity and semantic richness. TextAtlas5M contains 5 million generated and collected images across diverse data types, enabling comprehensive evaluation of large-scale generative models. We further curate 4,000 human-improved test cases (TextAtlasEval) spanning four domains, forming one of the most extensive benchmarks for text rendering. Evaluations show that TextAtlasEval poses substantial challenges even for state-of-the-art proprietary models (e.g., GPT-4o), with significantly larger gaps observed for open-source models. Training on TextAtlas5M consistently improves text rendering for both diffusion-based and autoregressive models, demonstrating its effectiveness for advancing text-rich image generation.

## 1. Introduction

Text-conditioned image generation is processing longer texts, with a growth from 77 tokens in Dall-E (Ramesh et al., 2021a) to 300 in PixArt-$\alpha$ (Chen et al., 2023c), and recently achieving 2,000-token capacity in autoregressive models (Team, 2024). In this regard, generating comprehensive and controllable images with longer text input, such as images with complex layout or dense text, is considered a promising testbed.

---

[*]Equal contribution  [1]Central South University  [2]North University of China  [3]Microsoft Research  [4]National University of Singapore. Correspondence to: Fuwei Zhang <20240048@nuc.edu.cn>.

*Proceedings of the 43$^{rd}$ International Conference on Machine Learning*, Seoul, South Korea. PMLR 306, 2026. Copyright 2026 by the author(s).

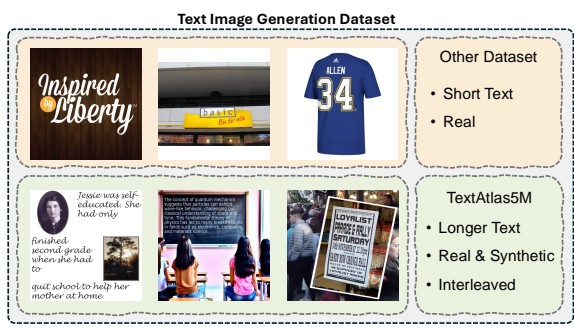

*Figure 1.* **Comparison of TextAtlas5M with previous datasets.** TextAtlas5M includes more diverse and complex long-form text than prior short-text datasets (Schuhmann et al., 2022; Tuo et al., 2023; Chen et al., 2024a).

Text plays a central role in image generation, serving as one of the most pervasive elements in visual communication through news, books, advertisements, and more. For instance, over 50% of images in LAION-2B (Schuhmann et al., 2022) contain text (Lin et al., 2025). However, despite the increasing prevalence of dense text in real-world scenarios, state-of-the-art models such as LlamaGen (Sun et al., 2024), Chameleon (Team, 2024), and TextDiffuser2 (Chen et al., 2023a) struggle with tasks requiring the generation of long and complex text.

This limitation stems from the reliance on existing text-rich image datasets like Marion10M and AnyWords3M (Tuo et al., 2023; Chen et al., 2023a), which focus on short and simple text, failing to meet the demand for handling longer and more intricate inputs. Such limitations are particularly evident in practical scenarios, ranging from advertising layouts that require seamless brand messaging integration to infographics that demand precise synchronization of text and visuals, as shown by the Notice Board in Figure 1.

To address these challenges, we introduce TextAtlas5M, a comprehensive dataset designed to advance and evaluate text-to-image generation models, with a particular focus on generating dense-text images. As illustrated in Figure 1 and Table 1, TextAtlas5M stands out in several key ways compared to previous text-rich datasets. Unlike earlier datasets, which primarily focus on short and simple text, TextAtlas5M includes a diverse and complex range of

*Table 1.* **Dataset Comparison with Existing Text-Rich Image Generation Datasets**. The last two columns detail the sources of automatically generated labels, while the final column presents the average text token length derived from OCR applied to the images.

| Dataset Name | Samples | Annotations | Domain | Labels | Token Length |
|---|---|---|---|---|---|
| TextCaps (Sidorov et al., 2020) | 28K | Caption | Real Image | Human | 26.36 |
| SynthText (Gupta et al., 2016) | 0.8M | OCR | Synthetic Image | Auto | 13.75 |
| Marion10M (Chen et al., 2024a) | 10M | Caption+OCR | Real Image | Auto | 16.13 |
| AnyWords3M (Tuo et al., 2023) | 3M | Caption+OCR | Real Image | Auto | 9.92 |
| RenderedText (Wendler, 2024) | 12M | Text | Synthetic Image | Auto | 21.21 |
| TextAtlas5M | 5M | Caption+OCR+Text | Real&Synthetic Image | Auto&Human | **148.82** |

data. It spans from interleaved documents(documents containing interleaved text and visual elements) and synthetic data to real-world images containing dense text, offering a more varied and challenging set of examples. Moreover, our dataset contains longer text captions that pose additional challenges for models, and incorporates human annotations on difficult cases to ensure a rigorous evaluation.

The synthetic subset progresses through three levels of complexity, starting with simple text on clean backgrounds. It then advances to interleaved data, blending text with visual elements, and culminates in synthetic natural images, where realistic scenes integrate seamlessly with text. The real image subset captures diverse, real-world dense-text scenarios. It includes filtered samples from datasets like AnyText (Tuo et al., 2023) and TextDiffuser (Chen et al., 2024a), detailed descriptions from PowerPoint slides, book covers, and academic PDF papers. To enrich diversity, we also gather dense-text images guided by predefined topics from CommonCrawl (Common Crawl, 2025) and LAION-5B (Schuhmann et al., 2022). To assess the capability of model in dense text image generation, we introduce a dedicated test set, TextAtlasEval, designed for comprehensive evaluation. This test set spans four distinct data types, ensuring diversity across domains and enhancing the relevance of TextAtlas5M for real-world applications. Our contributions are threefold:

① We introduce TextAtlas5M, *the first large-scale dataset specifically designed for long-text image generation*, which combines synthetic data with diverse real-world images covering dense-text scenarios across multiple domains.

② We introduce TextAtlas5M, a large-scale dataset of 5 million images specifically designed for dense text rendering, spanning from synthetic documents to real-world text-rich images with complex layouts and semantic structures

③ Our comprehensive evaluation reveals that even state-of-the-art models struggle with long-text generation. However, fine-tuning on TextAtlas5M consistently improves performance for both diffusion and autoregressive architectures, validating its utility as a training resource.

## 2. Related Works

**Text-conditioned Image Synthesis:** Generative modeling have prominently featured diffusion-based (Song & Ermon, 2019; Ho et al., 2020; Ho & Salimans, 2022) and autoregressive-based (Ramesh et al., 2021b; Esser et al., 2021a) frameworks. Diffusion models, such as DALL-E (Ramesh et al., 2021a) and Parti (Yu et al., 2022), produce high-fidelity outputs through an iterative refinement process but are limited by slow inference speeds. While autoregressive (AR) models (Lu et al., 2024; Sun et al., 2023; Zhan et al., 2024; Sun et al., 2024) model images as sequential token streams by using vector quantization (Van Den Oord et al., 2017) to discretize raw pixel data into tokens, which balances efficiency and sample quality, making AR modeling increasingly popular.

Despite significant progress, current methods still face challenges in generating dense, stylized text within images while maintaining high precision and aesthetic coherence. Our approach bridges this gap by building a carefully curated, text-rich dataset to enhance the accuracy and variety, even when conditioned on complex, lengthy prompts.

**Text-Image Pair Datasets for Generation:** MS-COCO (Lin et al., 2014) and TextCaps (Sidorov et al., 2020) are widely used image-text pair benchmarks. MS-COCO features descriptive annotations and TextCaps adds more contextually rich captions. Recently, CC3M (Changpinyo et al., 2021) and LAION (Schuhmann et al., 2022) further emphasize large-scale data sourced from the Web, which have been instrumental in training text-conditioned image generation models. However, both primarily cater to short or moderately long captions, limiting their suitability for tasks involving long text content. More recent efforts, Marion10M (Chen et al., 2024a) and Any-Words3M (Tuo et al., 2023), aim to diversify text inputs but often lack high-quality annotations or precise alignment, prioritizing visual scenes over accurate textual rendering. In parallel, several evaluation benchmarks, such as LongText-Bench (Geng et al., 2025), CVTG-2K (Du et al., 2025) and OneIG-Bench (Chang et al., 2025), have been proposed to access text-image alignment in generative models. This shows the growing interest of community in complex text-to-image scenarios. To bridge these gaps,

*we introduce TextAtlas5M explicitly designed for generating images from extensive and structured text.* This dataset addresses key limitations of existing resources and enables progress in long text-to-image generation.

**Visual Text Rendering:** Rendering text accurately in images requires balancing textural correctness, visual quality, and contextual coherence. Prior work in text image synthesis is broadly categorized into two directions: structured methods (Tuo et al., 2023; Yang et al., 2024; Ma et al., 2023; Chen et al., 2024a; 2023b; Liu et al., 2022), which enforce layout guidelines to achieve precise text placement for design-oriented tasks (e.g., posters), and unconstrained approaches (Chen et al., 2024c) that prioritize flexibility for long-text generation (e.g., documents) without extra guidelines. Despite recent progress, existing methods still struggle with the precise rendering of extended text due to the lack of large-scale, high-quality dense-text datasets. To address this, we introduce a diverse text-rich dataset that facilitates accurate and flexible text generation.

## 3. Dataset Construction

The primary goal of our TextAtlas5M is to collect diverse scenes in daily life containing dense text. However, acquiring high-quality real-world world text-rich data is both expensive and time-consuming. To balance quality and scalability, we first construct *Synthetic Image Split* with widely-used topics, providing easier cases for model training and evaluation. Further, we collect *Real Image Split* from diverse sources, including PowerPoint presentations, documents, and visually appealing real-world images. By combining these tiers, TextAtlas5M provides a comprehensive and scalable resource for dense text rendering. Figure. 2 shows an overview of the dataset.

### 3.1. Synthetic Images Split

**CleanTextSynth:** We create a simple dataset of text-only images, without incorporating additional visual elements, using the interleaved Obelics (Laurençon et al., 2024), in which we randomly sample text sequences of length $L$. Using OCR Rendering (GbotHQ, 2023) for text rendering, we place sequences on white canvases with 8,700 diverse font types (e.g., Helvetica, Times New Roman). We introduce significant variation in font size, font color, rotation angles, and text alignment to approximate the diversity of real-world visual text while retaining control over label quality. This results in 2 million samples of clean, well-formatted text, ideal for foundational experiments.

**TextVisionBlend:** Interleaved data seamlessly blends visual and textual elements in formats like blogs, wikis, and newspapers. Inspired by this, we created a synthetic interleaved image-text dataset that enhances data organization

and contextual richness. We source high-quality image-text pairs from Obelics (Laurençon et al., 2024) and WIT (Srinivasan et al., 2021), then design random layouts to automatically combine them. Using PyMuPDF (Inc., 2025), we generate white-background images and parseable PDFs, ensuring structured interleaved content. From these PDFs, we extract detailed annotations, including bounding boxes, font styles, and sizes, enriching each sample. To enhance contextual richness, we used vision-language models Qwen2-VL (Wang et al., 2024) and BLIP (Li et al., 2022) to generate image captions, consolidating all annotations and captions into comprehensive sample files. This dataset captures the complexity of interleaved data and see Supplementary material for more details.

**StyledTextSynth:** Building on pure-text images and interleaved text-image scenes, we address more complex embedded text scenarios, such as billboards, to enhance dataset diversity. The overall pipeline is shown in Figure 3. Using GPT4o (OpenAI, 2024) as a world simulator, we identify 50 real-world text-integration scenes, refining them into 18 high-frequency topics (e.g., urban signage, product packaging). GPT4o generates scene descriptions, which serve as prompts for SD3.5 to create text-free images. We then identify suitable text placement areas, refining them with human annotations as needed. Next, LLMs like GPT4o and Llama3.1 (Dubey et al., 2024) generate contextually relevant text, which is rendered into designated regions, producing fully annotated images aligned with each topic. See Appendix G for details.

### 3.2. Real Images Split

**PPT2Details:** We first consider PowerPoint presentations, a widely used and text-rich format. SlideShare1M (Araujo & Girod, 2016) containing 1 million PowerPoint slides in an interleaved format, with most slides featuring dense text. To annotate this dataset, we utilize Qwen2-VL (Wang et al., 2024). The text prompt is given in Appendix G.3. Each slide is first converted into an image, and the model is then used to generate detailed descriptions of the text, images, and other elements, such as diagrams, tables, and vectors from this image. To ensure high-quality annotations, we filter out slides without text and those of low quality. After this process, we retain a total of 0.3 million high-quality samples, providing a rich resource for further modeling.

**PPT2Structed:** In addition to PPT2Details, we further access detailed slide elements with bounding box annotations for high-quality PowerPoint presentations. The AutoSlide-Gen dataset (Sefid et al., 2021) comprises 5,000 slides derived from scientific papers, where presentations are crafted to effectively convey research innovations. To build this dataset, we process each slide using PyMuPDF (Inc., 2025) to extract element bounding boxes and their corre-

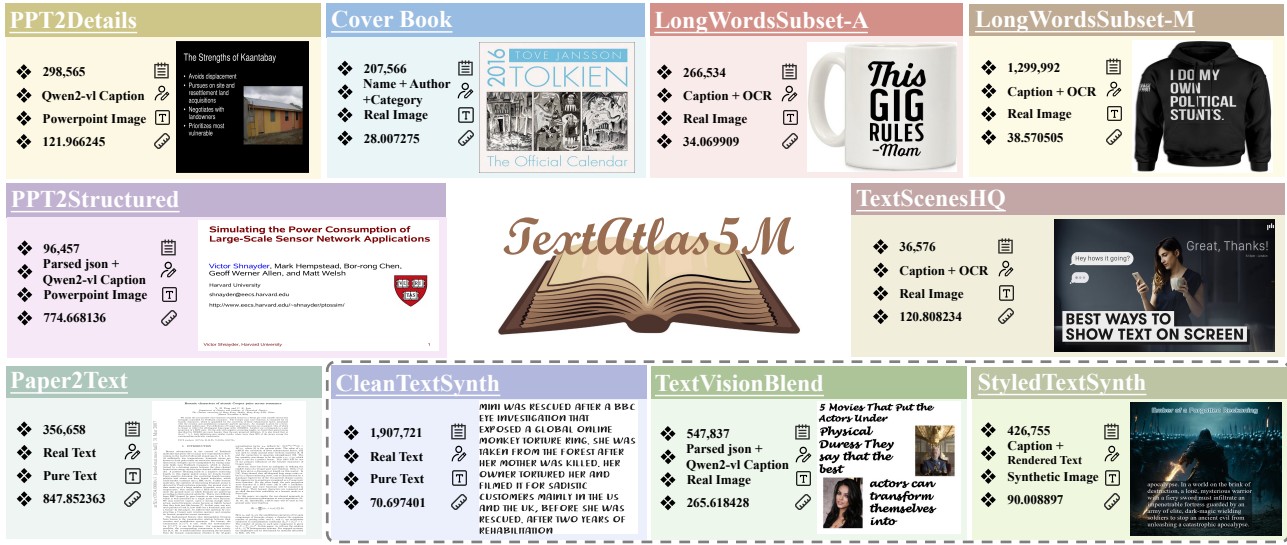

*Figure 2.* **Overview of TextAtlas5M Subsets.** TextAtlas5M comprises both Synthetic(boxed) and Real(unboxed) data splits. The synthetic split features three stages of increasing complexity, from clean text overlays to naturalistic text-image compositions. The real-world split is sourced from a diverse set of domains, capturing authentic dense-text scenarios for robust model evaluation.

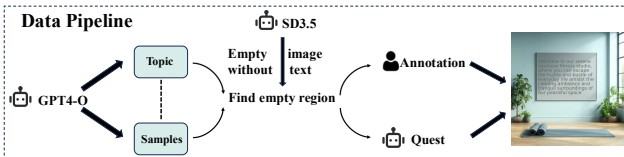

*Figure 3.* **StyledTextSynth Pipeline.** GPT-4o generates diverse text prompts, and a text-to-image model renders them into designated regions with seamless visual-text integration.

sponding text content. For slides containing images, we leverage the Qwen2-VL to generate descriptive captions. Text elements are preserved in their raw form to maintain accuracy and context. This process produces a structured dataset of 96,000 annotated samples, providing detailed elements along with their positional information.

**Paper2Text:** Another prominent text-rich scene is PDF documents, such as Arxiv papers. Using the Arxiv Paper dataset (arXiv.org submitters, 2024), we process each page by extracting its content with PyMuPDF. For this subset, we focus primarily on text information. Specifically, we retain attributes such as font color, size, and type. This approach enables the creation of detailed annotations containing descriptions of the text elements on each page.

**CoverBook:** We utilize the Cover Book dataset (Iwana et al., 2016), sourced from Amazon and Inc. marketplaces. This dataset comprises 207,572 books spanning 32 diverse categories, with each book providing a cover image, title, author, and category information. To create rich captions, we concatenate the title, author, category, and year information for each book.

**LongWordsSubset:** A straightforward approach to ob-

tain long-text samples is to filter existing text-rich image datasets. For this purpose, we use two widely adopted text rendering benchmarks: AnyWords3M (Tuo et al., 2023) and Marion10M (Chen et al., 2023a). Since most samples in these datasets contain short text, we apply a filtering process to select samples with longer words. The resulting subsets are named LongWordsSubset-A (from AnyWords3M) and LongWordsSubset-M (from Marion10M). To ensure data quality, we remove duplicates, repetitive patterns, and invalid text, retaining high-quality multilingual dataset samples. Detailed descriptions of the filtering process are shown in the Appendix H.

**TextScenesHQ:** To create a diverse and high-quality text-rich image dataset, we developed TextScenesHQ. Similar to StyledTextSynth, we use GPT4o as a world simulator to generate 26 predefined topics rich in text content. The overall pipeline is illustrated in Figure 4. The process begins with the retrieval images aligned with the specified topics from Common Crawl (Zhu et al., 2024). These images are then filtered using OCR-based filtering rules to select those containing long text. Images that do not meet this threshold undergo manual screening, during which we identify candidates for enhancement, such as adding text to advertisement backgrounds to enrich their visual complexity.

After cleaning, we annotate the images using advanced models Qwen2-VL and Intern-VL2 (Chen et al., 2024d). These models generate detailed textual descriptions and bounding boxes for detected text regions. To ensure annotation quality, we validate them through semantic similarity checks using LLM, ensuring consistency and relevance. For contrastive data and complex layout images, *we*

*Figure 4.* **TextScenesHQ Pipeline.** Data is filtered via OCR and manually refined to correct VLM-induced inconsistencies.

*incorporate human annotations to re-label the corresponding text to improve the data quality*. Finally, the curated images and their validated annotations are organized into a comprehensive dataset.

### 3.3. TextAtlasEval Generation

To rigorously evaluate dense-text image generation, we construct TextAtlasEval, a human-refined benchmark of 4,000 samples covering diverse domains and difficulty levels. We adopt a stratified sampling strategy: 1,000 samples each from StyledTextSynth, TextScenesHQ, and TextVisionBlend, representing synthetic, professional, and web-sourced interleaved scenarios.

For CleanTextSynth, we sample 1,000 text instances from Obelics and apply character-level truncation at 64, 128, 256, 512, and 1024 characters to **simulate increasing complexity**. For StyledTextSynth and TextScenesHQ, we sample uniformly across topics to ensure coverage. For TextVisionBlend, random sampling captures both controlled and organic scenes. All samples are manually refined to improve OCR accuracy and layout quality. Crucially, each image-text pair is *human-verified to guarantee that the image can be faithfully generated from its prompt*.

### 3.4. Unified Multimodal Data Construction

A major challenge in building TextAtlas5M is unifying heterogeneous annotations across sub-datasets. Most samples include either scene-level descriptions or rendered text, while some also provide fine-grained attributes like bounding boxes and font sizes. We define a unified representation combining scene context ($S$) and rendered text content ($T$), which forms the basis for long-text image generation.

To generate natural and coherent inputs, we leverage LLMs with adaptive prompting strategies tailored to each subset. For example, in LongWordsSubset, the LLM merges $S$ and $T$ into multiple diverse formulations. In StyledTextSynth, we use Qwen2-VL to generate scene captions with inserted placeholders (e.g., *"A cozy classroom with a blackboard displaying <>."*) to enable controllable text

rendering. More strategies are detailed in the Appendix G.

## 4. Analysis of TextAtlas5M

In this section, we first analyze the high-level statistics of our TextAtlas5M. We then analyze the topic distribution and qualitative properties of TextAtlas5M.

**Perplexity analysis:** We utilized the pre-trained Llama-2-7B (Touvron et al., 2023) to calculate perplexity scores for 10,000 documents from each dataset. Lower perplexity scores suggest a stronger resemblance to the types of text corpus used for Llama-2, including Wikipedia and other high-quality sources. Figure 6(a) presents the distributions of these scores. Our findings show that CleanTextSynth has significantly lower average perplexity than LongWordsSubset-A/M.

Additionally, the distribution of TextVisionBlend also aligns closely with that of the datasets used for Llama 2 training. We also observe that the text quality in synthetic datasets, such as CleanTextSynth, is significantly higher than that of real-image subsets like TextScenesHQ.

**Topic Analysis in StyledTextSynth and TextScenesHQ:** As illustrated in Figure 5, we present an analysis of the topic distribution in the StyledTextSynth and TextScenesHQ datasets. We additionally highlight the broader topic coverage in real images and find:

*i.* Our dataset encompasses a wide variety of text-rich topics, such as weather reports, banners, TV shopping ads, and monitor displays. This diversity is crucial for maintaining the richness and generalizability of the samples.

*ii.* By leveraging LLMs as world simulators to generate topics, we ensure that most topics are consistent across real and synthetic images, effectively bridging the gap between these two data sources.

*iii.* Some topics generated by the LLM are not suitable for rendering, either due to inherent complexities or limitations in the synthetic generation process. These topics are excluded to maintain the quality and feasibility of the dataset.

**Visual-Linguistic Similarity Assessment:** Our dataset primarily consists of English and Chinese, with other languages comprising only 0.3%. We assess image-text alignment using CLIP similarity, where text serves as the query and images as candidates. We adopt the ViT-B/32 pre-trained on LAION-2B (OpenCLIP Contributors, 2022). As shown in Figure 6(b), LongWordsSubset-A, -M, and Cover Book yield higher CLIP scores, likely due to the presence of image captions that enhance alignment. In contrast, interleaved data shows lower scores, highlighting a challenge

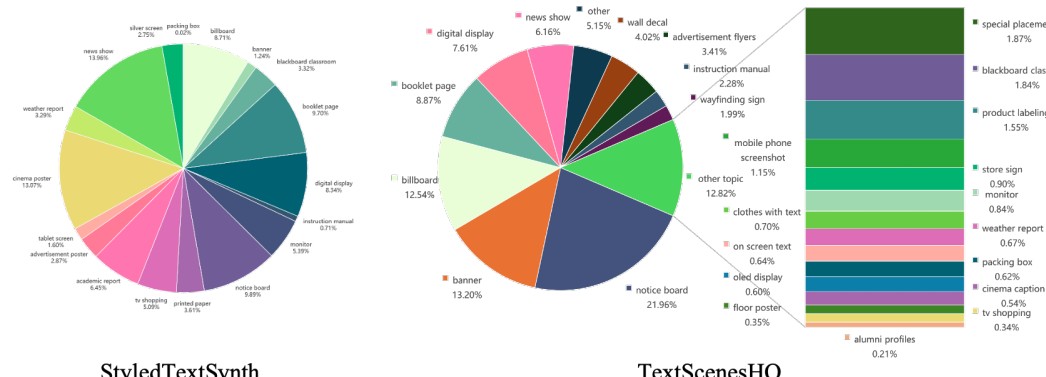

StyledTextSynth        TextScenesHQ

*Figure 5.* **Topic distribution in StyledTextSynth and TextScenesHQ subset,** showcasing a diverse range of text-rich topics. Styled-TextSynth includes carefully selected 18 topics, while TextScenesHQ ultimately contains 26 distinct topics. These topics are generated using GPT-4o as a **world simulator** and then filtered by humans to eliminate overlap while ensuring diversity.

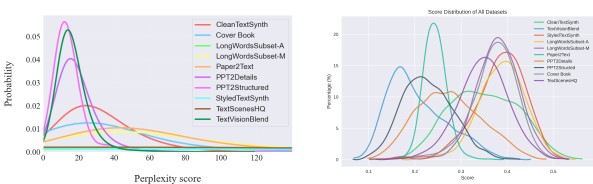

**(a) Perplexity Distribution.**     **(b) CLIP Score Distribution.**

*Figure 6.* **Linguistic and Visual Quality of TextAtlas5M.** (a) shows the perplexity distribution of text content, assessing linguistic quality. (b) presents visual-text alignment across subsets.

for CLIP in parsing visually entangled text layouts. Interestingly, Arxiv Paper scores higher than interleaved subsets, suggesting that CLIP retains partial OCR-like capabilities for document-style inputs.

## 5. Long Text Image Generation Evaluation

To evaluate our dataset for long-text image generation, we benchmark both autoregressive (AR) and diffusion models. Given the limited context capacity of diffusion models, we mainly focus on AR architectures better suited for long-form reasoning. Concretely, we fine-tune the Chameleon-7B (Team, 2024) (with Lumina-mgpt (Liu et al., 2024) decoder) at $512 \times 512$ resolution and Janus Pro 1B at $1024 \times 1024$. For comparison, we also train diffusion model PixArt$\alpha$ (Chen et al., 2023c).

We benchmark our fine-tuned models against eight state-of-the-art text-to-image generation systems, including open-source baselines (AnyText (Tuo et al., 2023), PixArt-$\Sigma$ (Chen et al., 2024b), TextDiffuser-2 (Chen et al., 2023a), Infinity (Han et al., 2024)) and commercial models (DALL·E 3 (Betker et al., 2023), SD 3.5 Large (Esser et al., 2024), QWEN Image (Wu et al., 2025). We also include closed-source models: GPT-4o (August 2025), Grok3 (May 2025) (xAI, 2025) and Nano-Banana (Septem-

ber 2025) (DeepMind, 2025). For evaluation, we compute image-text similarity using CLIP-ViT-B/32, where higher scores indicate stronger alignment. For OCR-based metrics, we use PaddleOCR (PaddleOCR Contributors, 2020) to extract generated text and compare it to ground truth. We report word-level accuracy, F1 score, and character error rate (CER), allowing up to 80% mismatch tolerance in word-level evaluation.

**Text-to-Image Generation Visualization:** To provide a clearer understanding, we present visual comparisons in Figure 7. Our observations highlight that previous open-source text-to-image generation methods beyond SD3.5 Large struggle with rendering dense text. For instance, AnyText renders only a handful of words, while Text Diffuser 2 captures only part of the text. In contrast, GPT-4 demonstrates the best performance in text rendering. These results underscore that dense-text image generation remains a challenging task for current models. Furthermore, we show the base model finetuned on our Dataset-Name in Figure 8. It is clear that base model often fail to understand text prompt without text rendering ability. The model trained with our dataset highly improving text rendering ability over all subsets even very hard real images.

**Performance Comparison on All Subsets.** Table 2 shows quantitative results for TextAtlasEval:

*i.* Models pretrained on our dataset exhibit significant improvements across all evaluation subsets, validating the effectiveness of long-text-focused data.

*ii.* CleanTextSynth and TextVisionBlend are relatively easier to learn: for instance, Lumina-mgpt and Pixart surpasses GPT-4o in CER on CleanTextSynth, indicating strong capabilities in pure text rendering. Meanwhile, lower CLIP alignment on TextVisionBlend suggests that its synthetic interleaved format remains underrepresented in training of CLIP, posing challenges for existing vision-

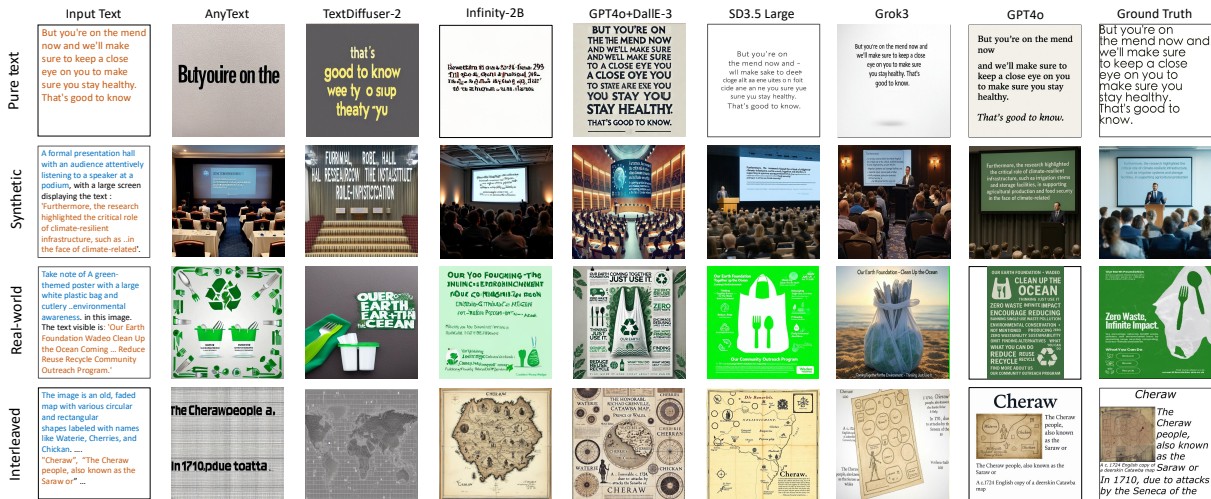

*Figure 7.* **Long text image generation remain a challenge for existing model.** Existing public methods, such as AnyText (Tuo et al., 2023) and Text Diffuser2 (Chen et al., 2023a), are capable of rendering short text but struggle with longer sequences. In contrast, GPT4o (OpenAI, 2024) and SD3.5 Large (Esser et al., 2024) show superior performance, though they still produce inaccuracies such as duplicated words or missing letters when handling extended text. For interleaved documents, all methods perform poorly due to their lack of layout planning capabilities. The yellow highlights text, while the blue indicates scene descriptions.

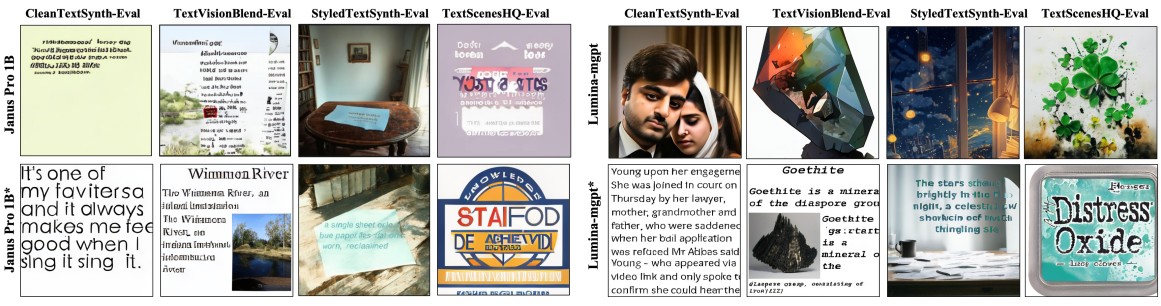

*Figure 8.* **Both Janus-Pro and Lumina-mGPT exhibit significant improvements in text rendering performance when fine-tuned on our TextAtlas5M**. Notably, while Janus-Pro has baseline support for generating text-rich images, Lumina-mGPT lacks inherent text rendering capabilities.

language models.

*iii.* In StyledTextSynth, *small font sizes make resolution a critical factor, where the tokenizer becomes a performance bottleneck.* For example, Janus Pro 1B outperform Lumina-mgpt 7B with higher resolution.

*iv.* TextScenesHQ remains the most challenging subset due to its diverse topics and complex layouts, requiring *both accurate prompt understanding and structured generation*. Even under such complexity, our pretrained model achieves a 3.7% gain in OCR accuracy, underscoring the value of diversity data for enhancing long-text image generation.

*v.* TextDiffuser2 is pretrained on Marion10M (Chen et al., 2023a), which is larger in scale than our TextAtlas5M. Nevertheless, both AR and diffusion models trained on our dataset outperform TextDiffuser2 by a clear margin. This highlights that *TextAtlas5M fills a critical gap by providing realistic, layout-rich, and semantically diverse text-image*

*pairs missing from existing benchmarks*.

**Layout-Aware Metric for Evaluation.** As an exploratory step toward better layout evaluation, inspired by object counting in GENEVAL (Ghosh et al., 2023), we introduce a layout-aware metric based on text bounding box counting. Our key insight is that while it is challenging to define what constitutes a "correct" layout, a reasonable generated layout should at least preserve the number of text regions. That is, the generated image should roughly match the source in terms of how many distinct text areas appear. This is straightforward to implement since we already have text region bounding boxes and an off-the-shelf OCR detector. We define the **IOU Count** metric as the percentage of images where the number of OCR-detected boxes matches the ground truth. The comparison is shown in Table 3. We observe that *the model trained on TextAtlas5M significantly improves the match rate, indicating that it has learned more accurate layout planning*.

*Table 2.* **Evaluation on TextAtlasEval across all four subsets.** Metrics include CLIP Score (CS) ↑, OCR Accuracy (Acc.) ↑, F1 Score (F1.) ↑, and Character Error Rate (CER) ↓. Gray rows denote closed-source models. Best and second-best values are highlighted in green and pink, respectively.To provide a fair comparison, the best performance among closed-source models is additionally highlighted in bold. † means model fine-tuned on our TextAtlas5M. Janus Pro 1B use $1024 \times 1024$ resolution and Lumina-mgpt 7B adopt $512 \times 512$ resolution. GPT4o, Dall-E 3 and SD-3.5 Large use $1024 \times 1024$ resolution and Grok 3 use $1024 \times 768$ resolution.

| Method | CleanTextSynth | | | | TextVisionBlend | | | | StyledTextSynth | | | | TextScenesHQ | | | |
|---|---|---|---|---|---|---|---|---|---|---|---|---|---|---|---|---|
| | CS | Acc | F1 | CER | CS | Acc | F1 | CER | CS | Acc | F1 | CER | CS | Acc | F1 | CER |
| AnyText | 0.21 | 0.18 | 0.34 | 0.99 | - | - | - | - | 0.25 | 0.35 | 0.66 | 0.98 | 0.22 | 0.42 | 0.81 | 0.95 |
| TextDiffuser-2 | 0.23 | 1.41 | 2.66 | 0.98 | - | - | - | - | 0.25 | 0.76 | 1.46 | 0.99 | 0.23 | 0.66 | 1.25 | 0.96 |
| PixArt-Σ | 0.24 | 0.69 | 1.15 | 0.92 | 0.19 | 2.40 | 1.57 | 0.83 | 0.28 | 0.42 | 0.62 | 0.90 | 0.23 | 0.34 | 0.53 | 0.91 |
| Infinity-2B | 0.24 | 0.11 | 1.93 | 0.88 | 0.20 | 2.98 | 3.44 | 0.83 | 0.27 | 0.80 | 1.42 | 0.93 | 0.23 | 1.06 | 1.74 | 0.88 |
| SD-3.5 Large | 0.28 | 12.0 | 18.2 | 0.84 | 0.18 | 14.55 | 16.25 | 0.88 | 0.28 | 27.21 | 33.86 | 0.73 | 0.24 | 19.03 | 24.45 | 0.73 |
| Qwen Image | - | - | - | - | 0.18 | 81.02 | 58.65 | 0.57 | 0.30 | 66.20 | 73.92 | 0.35 | 0.33 | 71.82 | 68.70 | 0.34 |
| Janus Pro 1B(1024) | 0.23 | 0.27 | 0.52 | 0.96 | 0.22 | 0.76 | 1.31 | 0.94 | 0.27 | 0.06 | 0.12 | 0.98 | 0.27 | 0.25 | 0.48 | 0.98 |
| Janus Pro 1B(1024)† | 0.27 | 5.23 | 8.97 | 0.65 | 0.18 | 13.61 | 17.55 | 0.61 | 0.28 | 12.75 | 17.37 | 0.75 | – | – | – | – |
| Lumina-mgpt7B(512) | 0.23 | 0.12 | 0.24 | 0.98 | 0.19 | 0.00 | 0.00 | 0.94 | 0.27 | 0.09 | 0.17 | 0.98 | 0.29 | 0.19 | 0.30 | 0.93 |
| Lumina-mgpt7B(512)† | 0.27 | 32.90 | 45.93 | 0.35 | 0.16 | 56.61 | 60.18 | 0.26 | 0.27 | 5.43 | 6.71 | 0.71 | 0.27 | 3.83 | 4.65 | 0.78 |
| Lumina-mgpt7B(1024) | 0.24 | 0.32 | 0.45 | 0.97 | 0.18 | 0.03 | 0.05 | 0.95 | 0.27 | 0.13 | 0.21 | 0.97 | 0.28 | 0.22 | 0.35 | 0.92 |
| Lumina-mgpt7B(1024)† | 0.29 | 48.55 | 59.67 | 0.31 | 0.17 | 65.34 | 70.32 | 0.22 | 0.29 | 27.93 | 20.20 | 0.49 | 0.27 | 6.44 | 9.34 | 0.79 |
| Pixart-α 0.6B(512) | 0.25 | 0.23 | 0.42 | 0.98 | 0.17 | 0.04 | 0.09 | 0.96 | 0.28 | 0.05 | 0.12 | 0.98 | 0.29 | 0.13 | 0.28 | 0.94 |
| Pixart-α 0.6B(512)† | 0.28 | 39.14 | 50.24 | 0.33 | 0.16 | 58.21 | 54.32 | 0.27 | 0.29 | 22.10 | 25.14 | 0.63 | 0.30 | 3.63 | 4.30 | 0.84 |
| Grok3 | 0.28 | 31.08 | 40.81 | 0.44 | 0.17 | 41.54 | 44.22 | 0.57 | 0.29 | 15.82 | 21.4 | 0.73 | 0.32 | 35.07 | 37.94 | 0.57 |
| DALL·E 3 | – | – | – | – | 0.19 | 8.38 | 7.94 | 0.93 | 0.29 | 30.58 | 38.25 | 0.78 | 0.34 | 69.26 | 51.63 | 0.67 |
| Nano Banana | 0.25 | 65.80 | 73.66 | 0.23 | 0.15 | 93.49 | 80.30 | 0.26 | 0.29 | 64.09 | 70.93 | 0.39 | 0.33 | 75.15 | 70.99 | 0.36 |
| GPT-4o | 0.27 | 60.69 | 74.44 | 0.36 | 0.15 | 91.78 | 82.07 | 0.15 | 0.30 | 77.47 | 80.76 | 0.21 | 0.33 | 82.88 | 78.68 | 0.32 |

*Table 3.* **Strict IOU Count Comparison on TextAtlasEval**. †: Model fine-tuned on TextAtlas5M.

| Dataset | Model | Average Detected Boxes | | Match Rate (%) | |
|---|---|---|---|---|---|
| | | Base | Fine-tuned† | Base | Fine-tuned† |
| TextVisionBlend | Pixart-α | 1.9 | 2.9 | 25.3 | 75.8 (+50.5%) |
| TextVisionBlend | Lumina-mgpt7B | 1.6 | 3.2 | 29.5 | 84.3 (+54.8%) |
| TextVisionBlend | GPT4o | 3.4 | – | 96.7 | – |
| TextScenesHQ | Pixart-α | 2.8 | 4.7 | 6.5 | 10.9 (+4.4%) |
| TextScenesHQ | Lumina-mgpt7B | 1.4 | 4.8 | 5.3 | 12.6 (+7.3%) |
| TextScenesHQ | GPT4o | 6.2 | – | 37.5 | – |

**Performance on Other Longtext Benchmark.** To further demonstrate the generalizability of our dataset, we evaluate both the trained autoregressive model (Lumina-mgpt) and diffusion model (Pixart-α) on two widely-used text-image benchmarks, LongText-Bench and CVTG-2K. The results(Table 4) reveal significant performance improvements of models fine-tuned on our data compared to their base versions. Specifically, Pixart-α achieves a substantial boost in the Text Score on LongText-Bench from 0.14 to 24.28, and improves Word Accuracy on CVTG-2K from 0.02 to 31.30. Similarly, Lumina-mgpt demonstrates gains on both benchmarks. These improvements across different benchmarks show the effectiveness of our dataset.

**Comparison on Short-Text Benchmark** We conduct additional evaluation of Pixart-α 0.6B (512) on

*Table 4.* **Performance Comparison on Other Benchmark**. We report text-rendering and layout-related metrics on LongText-Bench and CVTG-2K. All metric values are multiplied by 100 for better readability. †: Model fine-tuned on TextAtlas5M.

| Model | LongText-Bench | CVTG-2K | | |
|---|---|---|---|---|
| | Text Score | Word Acc. | NED | VQAScore |
| Pixart-α | 0.14 | 0.02 | 13.56 | 60.23 |
| Pixart-α† | 24.28 | 31.30 | 61.79 | 85.60 |
| Lumina-mgpt | 0.14 | 0.12 | 13.19 | 62.00 |
| Lumina-mgpt† | 9.11 | 6.02 | 23.68 | 54.30 |

*LAIONEval4000* (Chen et al., 2024a), a short-text English benchmark adapted from TextDiffuser. Results are reported in Table 5.

We find that Pixart-α trained on TextAtlas5M achieves sig-

*Table 5.* **Short-text rendering comparison on LAIONEval4000**. Pixart-$\alpha$ trained on TextAtlas5M outperforms training on Marion10M and achieves competitive or superior results compared to TextDiffuser.

| Method | Dataset | CS | Acc | F1 | CER |
|---|---|---|---|---|---|
| Pixart-$\alpha$ 0.6B (512) | - | 0.28 | 22.47 | 29.34 | 0.74 |
| Pixart-$\alpha$ 0.6B (512)[*] | Marion10M | 0.29 | 52.37 | 68.42 | 0.41 |
| Pixart-$\alpha$ 0.6B (512)[†] | TextAtlas5M | 0.29 | **67.53** | **76.55** | **0.36** |
| TextDiffuser | Marion10M | 0.28 | 53.25 | 71.44 | 0.38 |

nificant gains over the Marion10M baseline and performs on par with or better than TextDiffuser, despite the latter being tailored to short-text data.

Overall, TextAtlas5M fills a critical gap by providing layout-rich and semantically diverse text-image pairs that are underrepresented in existing benchmarks. It enhances long-text rendering performance while also improving robustness in short-text scenarios, thereby broadening the applicability of text-to-image generation models.

## 6. Conclusion

In this paper, we introduce TextAtlas5M, a novel large-scale dataset specifically designed for long-text rendering, addressing a critical gap in existing resources for text-conditioned image generation. To demonstrate its utility, we construct a dedicated test set and evaluate several state-of-the-art models, revealing their limitations in handling dense and complex text layouts. The open release of a diverse and high-quality dataset like TextAtlas5M provides a valuable foundation for both training and evaluating future-generation models in this domain.

While TextAtlas5M covers a broad range of long-text rendering scenarios, it may not fully represent specialized or structurally complex domains such as legal documents, scientific papers with embedded formulas, or multilingual posters. Future work could address these gaps by incorporating more diverse formats. Moreover, due to computational constraints, we leave several promising directions for future exploration, including iterative dataset bootstrapping and generating multiple synthetic captions per image to expand the corpus and improve training dynamics.

## Acknowledgement

This project is supported by the National Natural Science Foundation of China under Grant No. 62502544 and the Basic Research Program of Shanxi Province, funded by the Science and Technology Department of Shanxi Province, under Grant No. 202403021222152.

## Impact Statement

This work addresses the critical challenge of rendering longform text in text-conditioned image generation, introducing TextAtlas5M as a novel dataset and benchmark for evaluating and improving models in this domain. By demonstrating the performance gaps of both propriatary and open-source models, this benchmark sets a new standard for evaluating future generation text-conditioned image-generation systems, driving innovation and accountability in the field.

The potential societal impacts of this work are significant. On the positive side, improving long-text rendering capabilities can enhance accessibility and usability in applications such as video content creation, digital media production, and educational tools, empowering users to generate textrich, coherent visual content from text inputs. However, we acknowledge ethical considerations, such as the potential misuse of this technology for generating deceptive or harmful content. These risks highlight the importance of responsible development and deployment of text-conditioned generation models.

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

## A. Experiment Details

### A.1. Setup

To evaluate the effectiveness of our dataset, TextAtlas5M, we train and assess three representative models: two autoregressive models—Lumina-mGPT (Liu et al., 2024) and Janus-Pro (Chen et al., 2025)—and one diffusion model, PixArt-$\alpha$. All models are trained from scratch on TextAtlas5M to ensure a fair comparison. Most of our experiments were conducted on V100 GPUs. An exception is the evaluation of the Infinity model, which requires FlashAttention— a feature not supported on V100. As a result, we ran the Infinity experiments on A5000 GPUs.

### A.2. Training Details

**Lumina-mgpt model training:** For Lumina-mgpt (Liu et al., 2024) Training, our training process uses the AdamW optimizer, with $\beta_1$ sets to 0.9 and $\beta_2$ to 0.95, with an $\epsilon = 1e-5$. We use a linear warm-up of 4000 steps with an exponential

decay schedule of the learning rate to 0. Additionally, we apply a weight decay of 0.1 and global gradient clipping at a threshold of 1.0. We use a dropout of 0.1 for training stability.

**Pixart-$\alpha$ Model Training:** Follow Pixart-$\alpha$ (Chen et al., 2023c), the backbone architecture is DiT-XL/2. In contrast to prior works that limit text token length to 77, the token limitation is 120 tokens to accommodate the denser, more descriptive captions curated in the PIXART-$\alpha$ dataset especially for long text rendering, thereby enabling finer-grained image conditioning.

To encode visual inputs, we use a pre-trained and frozen VAE from LDM (Esser et al., 2021b), resizing and center-cropping all images to a consistent resolution prior to encoding. To support flexible aspect ratios during generation, we incorporate the multi-aspect ratio augmentation strategy from SDXL (Rombach et al., 2022). Training is performed using the AdamW optimizerwith a weight decay of 0.03 and a fixed learning rate of 2e-5. The final model is trained over on a cluster of 16 NVIDIA V100 GPUs.

**Janus-pro 1B Model Training:** We trained the Janus-Pro 1B model using the AdamW optimizer with an initial learning rate of $\epsilon = 1e - 4$ and a weight decay of 0.1. The input image resolution was set to 1024×1024, which is higher than the model's original pretraining resolution of 384×384, in order to better support high-quality text rendering. The training was conducted for 4 epochs. To ensure training stability, we employed gradient accumulation with one step and applied global gradient clipping with a threshold of 1.0.

### A.3. Evaluation Details

In this section, we provide detailed evaluation settings and observations for each model benchmarked on *TextAtlas5M*. We include both diffusion-based and autoregressive models with text-to-image generation capabilities. Unless otherwise specified, all models are used in their publicly available form without additional fine-tuning.

**Anytext (Tuo et al., 2023):** We evaluate the publicly released version *v1.1.3* of Anytext. Following its original setting in the demo, we set the inference steps of DDIM sampler as 20 and directtly input the prompt of our data. For the layout image required by the model, we randomly select from the set of layout templates provided by the authors of Anytext.

**TextDiffuser2 (Chen et al., 2023a):** For TextDiffuser2, we use the fully fine-tuned version provided by the authors. We adopt the default generation parameters from the official demo, including a maximum text length of 77, granularity of 128, classifier-free guidance scale of 7.5, and 20 sampling steps.

**PixArt-$\Sigma$ (Chen et al., 2024b):** We evaluate the model PixArt-$\Sigma$ using the *PixArt-Sigma-XL-2-1024-MS* version, which is a diffusion-based text-to-image generation model optimized for high-resolution rendering. All images are generated using the model's default configuration without any modifications.

**Infinity-2B (Han et al., 2024):** We evaluate Infinity using the *infinity-2b-reg* checkpoint, a 2B-parameter autoregressive model optimized for text-to-image generation. Classifier-free guidance is set to 4, following common practice for balancing fidelity and diversity. All experiments of Infinity-2B are conducted on NVIDIA A5000 GPUs.

## B. Qualitative Analysis of the Human-Refined Subset

### B.1. Analysis of StyledTextSynth

: We random select 154 images covering 15 topics, with no watermarks or NSFW content found. The OCR test results are as follows: In some topics, such as academic reports, the text in the images has a clear contrast with the background and a relatively large font size, resulting in a high OCR recognition rate. However, when different but still distinct font colors overlap, the OCR results become inaccurate. Erroneous fonts generated by SD3.5 also affect OCR performance. Moreover, environmental lighting in the images can interfere with OCR accuracy. In topics with poor rendering quality like booklet pages the OCR results tend to deteriorate.

### B.2. Analysis of TextScenesHQ:

We randomly sampled 200 images covering 23 topics, of which 4.0% found watermarks, but no NSFW images were detected. OCR recognition tests show that when the text is small or the contrast with the background is not obvious, the

recognition accuracy decreases; Quantitative analysis reveals 22.3% OCR accuracy degradation (from 89.4% to 67.1%) when text-background contrast drops below 30% RGB—a critical threshold for model robustness evaluation. In addition, some text is truncated due to blur or being located at the edge of the picture, affecting the recognition effect. For artistic words, the OCR recognition ability is poor, especially when the objects are designed as artistic words, they can hardly be correctly recognized, while artistic words similar to printed text have relatively good results.

Case studies show calligraphic text achieves 58.2% recognition rate versus 92.7% for standard fonts, exposing current models' typographic generalization limits. At the same time, among these 200 pictures, 14.0% of the pictures contain portraits (single portraits, group photos, advertising portraits and video covers), and 7.5% of the pictures contain pictures with logos.

## C. Additional Analysis on Dataset Significance

**Definition and Scope of Long Text.** Our notion of "long text" goes beyond simple sequence length. It also encompasses dense visual content, hierarchical layouts, and semantically rich structures. Subsets such as *TextVisionBlend*, *PPT2Structured*, and *TextScenesHQ* feature complex interleaved designs that pose substantially greater challenges than the short, clean captions prevalent in existing datasets. In this section, we provide further analysis to clarify the significance of our proposed dataset and its advantages over existing alternatives such as Marion10M (Chen et al., 2023a).

### C.1. Dataset Statistics

Figure 9 shows the proportion of each subset in our dataset, and Figure 10 reports the distribution of image-level word counts across subsets.

Notably, for structured and layout-heavy subsets such as *TextVisionBlend*, *PPT2Structured*, and *TextScenesHQ*, the effective token length is influenced not only by the number of words but also by the fragmentation of text across multiple regions. Such dense and interleaved layouts cause OCR to produce much longer token sequences than the raw word count alone would suggest, explaining the higher token lengths reported in Table 1.

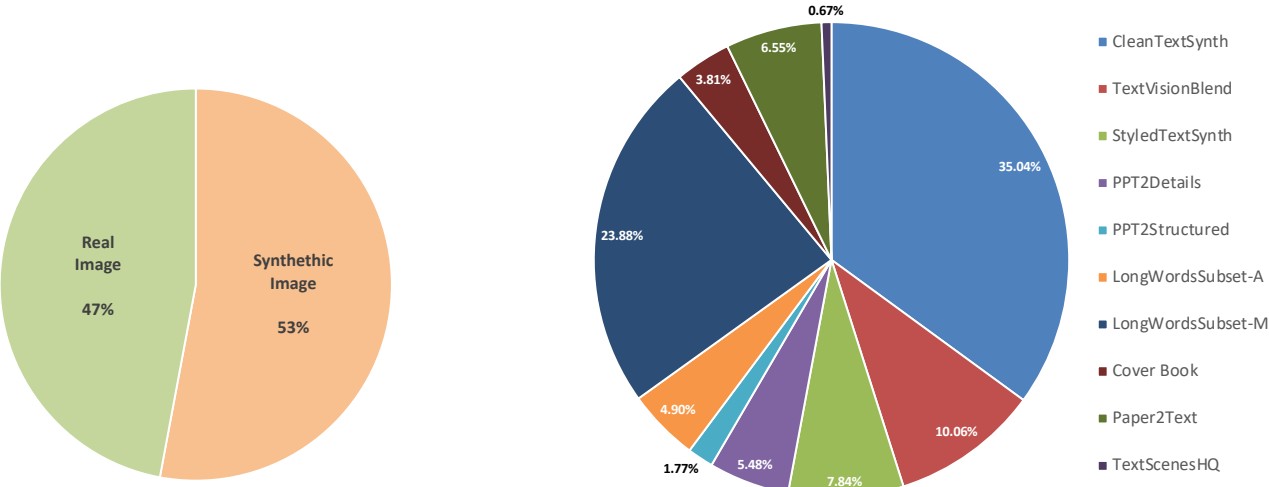

*Figure 9.* Dataset composition illustrated by the proportion of real and synthetic images (left) and the percentage distribution across all subsets (right).

### C.2. Aesthetic Analysis

We report aesthetic scores using the LION Aesthetics-Predictor (aesthetics-predictor-v1-vit-large-patch14), following the evaluation protocol of Chameleon. The generated images obtain a mean aesthetic score of 4.37.

This score should be interpreted in the context of the target image domain. Our benchmark focuses on text-heavy images with dense textual content and structured layouts, whereas the aesthetic predictor is primarily trained on natural images, such as photographs and artistic scenes. As a result, images containing large amounts of text are typically assigned lower

*Table 6.* Overview of *TextAtlas5M* Subsets: Data Splits, Annotations, and Average Token Lengths.

| Dataset Name | #Samples | Annotations | Type | Token Len. |
|---|---|---|---|---|
| **Synthetic Images** | | | | |
| CleanTextSynth | 1,907,721 | Real Text | Pure Text | 70.70 |
| TextVisionBlend | 547,837 | JSON + Qwen2-VL Caption | Pure Text | 265.62 |
| StyledTextSynth | 426,755 | Human + QWEN Caption | Synthetic Image | 90.00 |
| **Real Images** | | | | |
| PPT2Details | 298,565 | Qwen2-VL Caption | PowerPoint Image | 121.97 |
| PPT2Structured | 96,457 | JSON + Qwen2-VL Caption | PowerPoint Image | 774.67 |
| LongWordsSubset-A | 266,534 | Caption + OCR | Real Image | 38.57 |
| LongWordsSubset-M | 1,299,992 | Caption + OCR | Real Image | 34.07 |
| Cover Book | 207,566 | Name + Author + Category | Real Image | 28.01 |
| Paper2Text | 356,658 | PDF Text | Pure Text | 847.85 |
| TextScenesHQ | 36,576 | Human + LLaMA + Qwen + GPT4o | Real Image | 120.81 |
| **In Total** | | | | |
| **TextAtlas5M** | **~5M** | — | — | **148.82** |

aesthetic scores by such predictors, independent of text correctness or structural quality.

Therefore, the reported aesthetic score mainly reflects a domain mismatch between the evaluation model and text-centric image distributions, rather than overall generation quality. Our primary objective is to improve text fidelity and structural alignment, and we include this metric for completeness.

### C.3. Addressing Synthetic Data Bias and Generalization

While our dataset includes synthetic content, we also incorporate a significant portion of natural images, constituting approximately 47% of the dataset. Specifically, the subsets CoverBook, LongWordsSubset, PPT2Details, PPT2Structured, and TextScenesHQ are derived from real-world data. For challenging subsets such as StyledTextSynth and TextScenesHQ, we introduce human-in-the-loop annotation to improve alignment quality and ensure higher annotation reliability. Additionally, all samples in the TextAtlasEval benchmark are human-checked and annotated.

To mitigate overfitting to synthetic patterns and enhance generalization, we adopt a mixed training strategy that combines synthetic data with real-world subsets from TextAtlas5M. This approach leads to improved performance, particularly on complex document layouts and instruction-following tasks. We treat the following subsets as real-world: PPT2Details, PPT2Structured, LongWordsSubset, CoverBook, and TextScenesHQ. The remaining subsets are categorized as synthetic.

To evaluate the effectiveness of mixed training, we fine-tune the Lumina-mGPT model under two settings: (1) trained on real-world subsets only, and (2) trained on both real and synthetic subsets. We report results on the challenging TextScenesHQ Eval benchmark, as shown in Table 7.

*Table 7.* Performance Comparison on TextScenesHQ Eval Benchmark

| Training Data | CLIP-Sim ↑ | Accuracy ↑ | F1 Score ↑ | CER ↓ |
|---|---|---|---|---|
| Real-Only | 0.27 | 5.32 | 7.31 | 0.82 |
| Mixed (Real + Syn) | 0.27 | 6.44 | 9.34 | 0.79 |

We observe that models like Lumina-mGPT and Pixart-$\alpha$ initially struggle to generalize beyond synthetic distributions. However, after approximately 20,000 steps of mixed training, both models exhibit stronger layout understanding and more stable generation behavior. These results suggest that synthetic pretraining, when complemented with real-world layout data, can significantly improve downstream performance on realistic and structurally complex text-image benchmarks. Together, these components reduce distributional bias and help bridge the gap between synthetic and real-world settings.

## D. Text Rendering Ability Explorison

### D.1. Impact of Long Sequences on OCR Performance

To better understand the relationship between text length and text rendering quality, we conduct an evaluation on the CleanTextSynth split of our TextAtlasEval. This benchmark is specifically designed to isolate text rendering performance by removing background visual content—only pure text is present in each image. Figure 11 illustrates how OCR accuracy

varies with increasing text length.

We derive the following key observations: Performance gains from dataset training: Both Janus-Pro-1B and Lumina-mGPT-7B show substantial improvements over their baseline versions when fine-tuned on our TextAtlas5M, highlighting the benefits of our data for text-centric generation tasks. Length sensitivity: As expected, OCR accuracy significantly decreases with longer text sequences, indicating a persistent challenge in maintaining rendering quality at scale. Competitive with GPT-4o: Our models outperform all open-source baselines and approach the performance of GPT-4o. Notably, in some cases, **our Character Error Rate (CER) is even lower than that of GPT-4o, further validating the effectiveness of our dataset in handling long-text scenarios**.

### D.2. Robustness Analysis on Short Sequences

We provide a detailed analysis of how model performance varies with short input text length from 2 to 64. Since the Clean-TextSynth evaluation set lacks short-text cases, we additionally curated 140 samples from the interleaved Obelics dataset and constructed text variants ranging from 2 to 64 tokens. Each version was rendered using five representative models, including both open-source and closed-source systems. We employed Qwen2-VL as the OCR engine and computed standard recognition accuracy.

*Table 8.* Recognition accuracy (%) as token length increases. [†] Model fine-tuned on TextAtlas5M.

| Token Length | AnyText | TextDiffuser2 | SD3.5 Large | Grok3 | GPT-4o | Lumina-mgpt[†] |
|---|---|---|---|---|---|---|
| 2 | 98.5 | 94.3 | 92.3 | 100.0 | 100.0 | 100.0 |
| 4 | 94.3 | 92.2 | 89.4 | 100.0 | 100.0 | 100.0 |
| 8 | 88.5 | 87.4 | 84.6 | 96.5 | 100.0 | 100.0 |
| 16 | 47.3 | 61.5 | 78.3 | 93.2 | 100.0 | 98.4 |
| 32 | 23.9 | 28.5 | 62.5 | 88.4 | 98.5 | 93.2 |
| 64 | 17.4 | 18.9 | 45.3 | 76.5 | 96.7 | 85.7 |

We observe: *i.* Performance of open-source models degrades sharply beyond 8–16 tokens. For example, TextDiffuser2 drops from 88.5% at 8 tokens to only 11.5% at 128 tokens. *ii.* In contrast, GPT-4o and Grok3 exhibit strong robustness, maintaining accuracy above 76% even for 64-token inputs.

**Conclusion.** This analysis highlights two critical insights: (i) most open-source models still struggle with rendering long or dense text, particularly beyond 10 tokens; and (ii) the primary cause is insufficient long-text supervision in prior datasets, which predominantly emphasize short, isolated words or captions.

### D.3. Visualization of Generated Samples

Figure 12 showcases representative samples generated by Lumina-mGPT after being trained on our TextAtlas5M. It is important to note that the **base model initially lacks any inherent text rendering capability**.

From the visualization, we observe the following: Significant improvement in text rendering across all data types, indicating effective adaptation through our dataset. Preservation of natural image generation quality, even in challenging scenarios such as realistic face synthesis—demonstrating that fine-tuning for text rendering does not compromise general visual fidelity. These results highlight the strength of our dataset in enhancing text understanding while retaining high-quality image generation.

## E. Autoregressive vs. Diffusion Models on Text Rendering

### E.1. Tokenizer Has a Major Influence

In our training and evaluation, we observed that using VQ-VAE as a vision tokenizer notably impacts the performance of autoregressive generative models. Unlike VAEs, VQ-VAE quantizes continuous encoder features into discrete codebook entries, which introduces non-negligible information loss. This makes it challenging to reconstruct fine-grained visual details—such as small objects or complex structures—especially under low-resolution settings. To further examine this effect, we compare VQ-VAE tokenizer from Janus-Pro (Chen et al., 2025) and VAE tokenizer from stable diffusion (Esser et al., 2024) reconstructions at a fixed resolution of 512×512. As shown in Figure 13, VQ-VAE struggles more with text

fidelity and structural sharpness, reinforcing the limitations of discrete tokenization in preserving detail.

We further illustrate the resolution-dependent behavior of the Janus Pro's VQ-VAE tokenizer in Figure 14. As resolution increases, the model becomes more capable of reconstructing textual and structural details. For instance, at 256×256, certificate text and product labels are barely recognizable, while at 1024×1024, the reconstructions closely resemble the ground truth. However, this comes with a trade-off: the number of tokens grows quadratically with resolution, significantly increasing the computational burden for AR models.

### E.2. Generation result comparison

In this section, we compare the text rendering capabilities of our diffusion-based model PixArt-$\alpha$ with the autoregressive model Lumina-mGPT, both trained on our proposed dataset, TextAtlas5M.

For the diffusion model, we evaluate two types of prompts: (1) "Generate an image of size $512 \times 512$ according to the following prompts: xxx", and (2) "A billboard outdoors with text: xxx".

The qualitative results are presented in Figure 15. Our key observations are as follows:

- **Detail rendering:** The diffusion model excels in visual fidelity, even at a resolution of $512 \times 512$. It produces well-formed and accurately aligned text. This advantage is largely attributed to its use of continuous token representations, which offer finer granularity than discrete tokens used in autoregressive models.

- **Textual coherence:** The autoregressive model demonstrates superior coherence in word sequence and correctness of generated content. However, its visual rendering is less precise—words often appear loosely structured or distorted. We attribute this to the nature of autoregressive decoding, which focuses on sequential word prediction rather than global image consistency.

- **Layout consistency:** While diffusion models render text sharply, they occasionally introduce hallucinated or irrelevant text, especially in complex layouts. This inconsistency reflects the model's weaker control over semantic alignment in dense text scenarios.

## F. Visualization

### F.1. Layout Planning Comparison

It is worth noting that although SD-3.5 (Esser et al., 2024) Large significantly outperforms PixArt (Chen et al., 2023b) and Infinity (Han et al., 2024) in OCR-related scores on the TextVisionBlend subset, its FID and CLIP scores are lower than those of the other two models. To better understand this phenomenon, we present two representative cases in Figure 17, analyzing the differences in model performance on this subset.

In the first row of Figure 17, SD-3.5 fails to capture the interleaved image layout and does not render text well, whereas both Infinity and PixArt follow the interleaved structure and white-background requirement, despite their poor text quality. This may explain SD-3.5's lower CLIP and FID scores. Meanwhile, in the second row, all three models exhibit interleaved characteristics, but only SD-3.5 generates relatively complete text in the image. This likely contributes to its strong OCR-related performance.

Overall, when generating images with complex requirements, SD-3.5 performs poorly in terms of image layout and certain specifications. We speculate that this may be related to the model's supported input text length. PixArt-Sigma can accommodate up to 300 text tokens, while Infinity, as an autoregressive generation model, supports even longer text inputs. A greater text input capacity may provide an advantage in understanding complex instructions.

### F.2. Comparison of Existing Models on TextAtlasEval

We present a comparative analysis of existing text-to-image generation models in Figure 16. Among all models, GPT-4o significantly outperforms others across all subsets of TextAtlasEval, demonstrating a substantial lead in both visual quality and text fidelity. Grok3 also shows strong performance, yet the gap between closed-source and open-source models remains considerable.

Notably, with the introduction of our benchmark dataset, TextAtlasEval, we observe a noticeable reduction in this performance gap—highlighting its effectiveness in driving progress and bridging disparities between different model families.

# G. Creation of the Synthetic Dataset

## G.1. Creation of StyledTextSynth

### G.1.1. TEXT PROMPT FOR STYLEDTEXTSYNTH IMAGE GENERATION

**General Prompt:** The core of the synthetic data involves utilizing text-conditioned image generation methods. For simple topics like billboards, we follow a "General Prompt" approach with the following guidelines:

- Provide a reasonable description of the billboard.

- Ensure the billboard faces the camera directly.

- The billboard should occupy at least one-third of the image.

- Incorporate a complex background for added detail.

- Keep the billboard's color consistent, with no additional context.

- Limit the total text on the billboard to fewer than 160 words.

- Ensure the billboard is visible, vehicle-related, and not overlapped by other objects.

We show an example in Table 9, with detail instruction the LLM generate reasonable scene description for image generation model.

*Table 9.* An example prompt and generated description for a silver screen image.

---

**GPT4o:**
I want to use model to generate some pictures of silver screen, so please give me some prompt follow these rules. Silver screen without any content and pure color but vertical , and looks like facing the camera but don't have any content, and the silver screen should take up 1/3 of the image, and have some specific complex background description, and the total word count should not exceed 160 words result is a paragraph.

**GPT4o res:**
A vertical silver screen, empty and pure in color, facing the camera directly. The screen occupies the lower third of the image, with a rugged mountain range in the background, peaks dusted with snow, and the sky painted in warm hues of a fading sunset.

---

**Topic with Human-Designed Seeds:** Certain topics, such as *studio* scenes, can involve highly complex setups. To address these challenges, we simplify the generation of text regions by incorporating human-designed seeds. Specifically, the instructions include a *general prompt* combined with a predefined, human-curated scene to serve as the seed, ensuring better coverage and control over these intricate scenarios.

- Weather Report: A host stands in a studio, with a display screen behind them showing weather conditions for various locations. In the foreground, a white rectangular box occupies one-third of the screen.

- TV Shopping: A host stands beside a table, holding a product in their hand, with an advertising board placed vertically next to them.

- Instruction Manual: An instruction manual is placed vertically, with objects associated with it positioned nearby.

**Template Topics with Fixed Text Region Position**: Some topics can be used as template to generate more samples, what is different is that the text appeared position is fixed. For these generate samples, we can simply generate more samples by simple text replace.

- News: Describe an image that could appear on television news. It can depict either positive or negative events, with diverse content. The total word count should be less than 160 words.

- Cinema Poster: Describe an image of a movie poster. It can be of any genre. The total word count should be less than 160 words.

*Table 10.* Rendered Text Generation for Specialized Scenes Using the GPT4o Model.

**GPT4o:** Please generate 50 texts about noticeboard, 40-80 words per text, the requirements are as follows: Any notification is fine. Only output text, and pay attention to the diversity of output, and do not have text with repeated meanings. The output format should be 1.text 2.text...

GPT4o res:
1.The monthly community cleanup event will take place this Saturday at 9 AM. Volunteers are encouraged to gather at the community center with gloves and trash bags. Let's work together to keep our neighborhood clean! 2.Attention students: The library will close an hour earlier than usual at 5 PM on Friday due to maintenance work. Please plan your study sessions accordingly. ... 50. The community board game night is happening this Friday at 7 PM in the activity room. Bring your favorite games!

*Table 11.* Rendered Text Generation for Specialized Scenes Using the Llama3.1 Model.

**Llama3.1:**
Please generate 100 texts about noticeboard, 40-80 words per text, the requirements are as follows: Any notification is fine. Only output text, and pay attention to the diversity of output, and do not have text with repeated meanings. The output format should be 1.text 2.text ...

Llama3.1 res:
1.The Student Union is hosting a Cultural Festival on August 10th to celebrate the diversity of our campus community. Come and experience the vibrant traditions, music, and food from around the world. There will be performances, workshops, and a marketplace featuring local vendors. All are welcome to attend. ... 100.The Neurodiversity Awareness Group invites you to a seminar on neuroinclusive education on Wednesday, November 15th, at 4:00 PM in the lecture hall. Learn about the latest research and strategies for supporting students with neurological differences.

G.1.2. UTILIZING LLMS FOR TEXT GENERATION IN SPECIFIC SCENES

After generating scene images without text, it is necessary to call upon an LLM to create contextually relevant sentences for rendering. To ensure realism in the generated outputs, we employ **Scene-Dependent Text Generation**. For general topics, such as noticeboards, we prompt the LLM to produce sentences based on the given topic. Examples of such outputs are shown in Table 10 and Table 11. The LLMs accurately generate text appropriate for the given scene.

**Visual-Dependent Scenes:** In some cases, the visual appearance of the scene is closely connected to the text and requires fine-grained visual understanding. For instance, a generated poster with rich visual elements may need text that complements its design. In such scenarios, we use LVM models that process both text and image inputs to produce reasonable outputs. An example of this process is shown in Table 12. By incorporate specific instruction, the model produce reasonable output.

*Table 12.* Rendered text generation for specialized scenes using the QWen2-VL model.

**QWen2-VL Synopsis:**
This is a picture about a movie poster. Please imagine a story for the picture based on the content of the picture, and then output a story outline. Just output the story outline, do not output other content. The story outline should be between 30-60 words.

QWen2-VL Synopsis res:
In a world on the brink of destruction, a lone, mysterious warrior with a fiery sword must infiltrate an impenetrable fortress guarded by an army of elite, dark-magic-wielding soldiers to stop an ancient evil from unleashing a catastrophic apocalypse.

QWen2-VL Title:
This is a picture about a movie poster. Please imagine a story for the picture based on the content of the picture, and then give the story a title. Only output the title, do not output other content, and the title should not exceed 6 words.

Qwen2-VL Title Result:
Ember of a Forgotten Reckoning

Final Transformation:

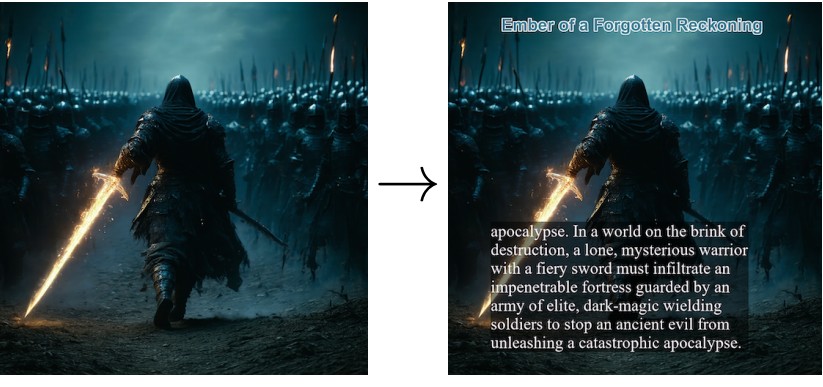

### G.1.3. HOW DO WE SELECT DATA TOPICS FOR RENDERING?

Originally we generate 50 topics that include dense text, one more question is how to filter these topics. With this in mind, we design the following filter rules:

1. **Avoid topics directly tied to font generation**, such as store signs, wayfinding signs, or on-screen text.

2. **Exclude topics where the renderable area is too small**, such as mobile phone screenshots.

3. **Avoid topics with unclear boundaries or artistic fonts that are hard to recognize**, such as neon signs.

4. **Prioritize topics with better rendering results in SD3.5 over similar ones**, e.g., choose "digital display" over "OLED display" or "banner" over "protest marches."

### G.1.4. TEXT DEDUPLICATION IN STYLEDTEXTSYNTH DATA

The text in the synthetic data is generated by Llama 3.1, GPT4o, and Qwen2VL. In most topics, there is basically no obvious disharmony between the generated images and the scenes, so we mainly use the text generated by Llama 3.1 and GPT4-o. For some scenes where the images and texts are highly correlated, VLM is needed to generate texts that match

the image content. Under the same or semantically similar topics, there may be semantic duplication. To this end, we semantically deduplicate the generated text based on the sentence-transformers library.

**Deduplication Process with Semantic Hashing** The deduplication process consists of the following steps:

1. **High-Dimensional Semantic Representation:** Obtain the high-dimensional semantic representation of the text.

2. **Dimensionality Reduction:** Map the high-dimensional semantic vector to a fixed low-dimensional space using random projection.

3. **Semantic Hash Generation:** Generate semantic hashes based on the projection results.

4. **Pairwise Similarity Comparison:** Use Hamming similarity to perform pairwise comparisons of the semantic hashes. A Hamming similarity threshold of 0.9 is applied to detect and remove semantically similar texts.

This structured approach ensures effective and accurate text deduplication while maintaining semantic integrity.

### G.1.5. MIDDLE-QUALITY SAMPLE FILTERING

To enhance the quality of the generated middle-quality samples, we apply a set of filtering rules to reject unsuitable samples. The primary criteria for rejection are as follows:

- Insufficient areas available for rendered text.

- Excessive similarity to other topics, reducing diversity.

- Difficulty rendering text in curved areas.

- Unrealistic or artificial appearance of the image.

- Challenges in identifying or defining bounding boxes.

- Presence of incorrect or irrelevant text.

- Poor recognition quality or unclear visual details.

Examples of rejected samples are illustrated in Figure 18.

### G.2. Gen Interleaved Data Benchmark

The process of generating interleaved data is divided into three main parts: data selection, PDF generation, and annotation generation.

### G.2.1. DATA SELECTION

We select data from WIT (Srinivasan et al., 2021) and OBELICS (Laurençon et al., 2024). The WIT dataset contains samples from Wikipedia, with each sample comprising an image and multiple associated text segments, such as titles, main text, subtitles, subtext, and image captions. From this dataset, we sample instances containing a single image and interleaved text. The OBELICS dataset consists of interleaved image-text documents sourced from web pages in the Common Crawl. Each OBELICS sample includes multiple images with their corresponding text segments. For our purposes, we sample data containing two to four images to maintain manageable image sizes. Finally, we sampled 69.8% data from the WIT and 30.2% data from the OBELICS, respectively.

### G.2.2. PDF GENERATION

After selecting the data, we use the PyMuPDF (Inc., 2025) library to generate parseable PDF files based on the sampled data. To accommodate the two types of data, we design different layout generation strategies according to their respective structures.

*Table 13.* Description generation prompt.

| Generate data template | |
|---|---|
| Prompt | "I have a scene description T and OCR text O. Please generate 200 unique combinations that naturally merge the description and OCR text into cohesive, longer paragraphs. Save the output in a plain text file." |

For both datasets, the layout strategy involves first randomly assigning image positions on the page, followed by allocating text boxes in a manner that optimally utilizes the remaining space. For the WIT dataset, since the text segments have predefined types (e.g., title, main text, subtitle), we impose additional constraints to ensure structural consistency. For instance, titles are placed at the top of the corresponding image to maintain semantic alignment. In contrast, for the OBELICS dataset, we adopt a simpler approach where the text boxes are sequentially assigned in a top-left to bottom-right order across the layout.

To implement the layouts, we use the `insert_htmlbox()` from PyMuPDF to insert images and text into the PDFs. The font for each sample is randomly selected to introduce variation. To further standardize the generated PDFs, we limit the text in each text box to a maximum of 50 words. Additionally, after generating the PDFs, we save a rendered image version of each page to serve as the corresponding image data for our dataset.

### G.2.3. ANNOTATION GENERATION

After generating parseable PDFs, we use the PyMuPDF to extract information such as the bounding boxes of text and images, as well as text font sizes and styles. Additionally, we utilize Qwen2-VL to generate captions for each image within the PDF. The prompt used for caption generation is: `"Generate the caption of the image, and the caption should be no more than 50 words."`

Finally, we obtain detailed information for the text, including its content and bounding boxes, along with the captions and bounding boxes of the images. By combining these elements based on the generated template, we produce the interleaved data annotations.

### G.3. Template Generation Details

To create templates for summarizing descriptions and text, we utilized an LLM to generate a total of 600 templates. The prompt used for this process is detailed in Table 13.

### G.4. Bounding Box Annotation and Detector Training for StyledTextSynth Sample

#### G.4.1. BOUNDING BOX GENERATION

1. **YOLO Results:** Based on the model's output, select the bounding box with the largest rectangular area when multiple results are present.

2. **Fine-tuned RT-DETR_R50VD Results (Packing Box):** Use the model's output to identify the bounding box. If multiple results are present, select the one with the smallest difference between width and height.

3. **RT-DETR_R50VD Results (Booklet Page):** Check if the output contains the label "book". If multiple results are present, choose the bounding box with the smallest width-to-height difference.

For all three methods above, the results are passed through SAM2 (Segment Anything Model v2) to refine recognition. The SAM2 output is converted into center points to serve as prompts, improving predictions for slanted surfaces.

#### G.4.2. DETECTOR TRAINING

**YOLO Training Process:**

1. For each topic (excluding template topics like Alumni Profile and News, as well as rejected topics), start with the

YOLOv11l initial weights and manually label about 1,000 failed detection samples. Use the following annotation methods:

(a) For slanted images, use the `labelme` tool to annotate with quadrilateral bounding boxes (four points).

(b) For upright images, annotate using the `Code` tool with two points (top-left and bottom-right).

Annotate areas of the image where the topic is unobstructed, and convert all annotations to YOLO format with the class label set to 0.

2. Train the model on the mixed annotated dataset for approximately 400 epochs (or more) to obtain initial weights.

3. Test the initial weights on each topic. A detection rate of at least 40% is considered usable for that topic.

4. For topics with low detection rates in Step 3, augment the manually labeled dataset for that topic with approximately 1,000 additional images. Apply transformations such as scaling, composition, flipping, and affine transformations. Combine the augmented data with the original manually labeled data (totaling approximately 2,000 images), and retrain the model for an additional 200 epochs.

5. Certain topics may share weights based on detection performance. For example:

   - **Billboard** and **TV Shopping** can share the same weights.
   - **Blackboard Classroom** and **Advertisement Poster** can share the same weights.

**Fine-Tuned RT-DETR_R50VD Training:**

1. Fine-tune the RT-DETR_R50VD model specifically for the **Packing Box** topic. Use 1,000 manually annotated packing box samples from Step 1.

2. Train the model for approximately 100 epochs.

### G.5. Text Rendering Details

G.5.1. Bbox Text Rendering:

After obtaining images generated by Stable Diffusion and images from CommonCrawl, which contain large fillable text areas (such as billboards, electronic screens, etc.), we use YOLO v11 and RT-DETR_r50vd to identify and label the fillable areas in the images. However, these detectors can only recognize rectangular areas, and the labeled fillable regions are often slightly larger than the actual fillable areas. Therefore, we further use SAM2, starting from the center point of the bounding box, to search for color-matching areas. This ensures that the new bounding boxes generated by SAM2 more accurately cover the fillable areas, breaking the traditional rectangular limitation and supporting the detection and filling of irregular quadrilaterals.

For text content generation, we use Llama-3.1-8B, GPT-4o, and Qwen2-VL-7B. Among them, Qwen2-VL-7B is mainly used for generating text related to cinema posters.

**Rectangular Bbox Text Rendering** For detected rectangular bounding boxes, we directly render text within the area. The font is randomly chosen from 10 common fonts, and the font size is automatically adjusted based on the bounding box size to fill the area as much as possible, ensuring both aesthetics and readability.

**Irregular Quadrilateral Bbox Text Rendering** For detected irregular quadrilateral bounding boxes, we first create a transparent layer and render the text on that layer. Then, we use a perspective transformation to adjust the transparent layer to match the irregular quadrilateral shape of the bounding box and finally composite it onto the original image, ensuring the text accurately fits the fillable area.

G.5.2. Template Rendering Method:

For images related to News Shows, Weather Reports, and Cinema Posters, where text usually appears in relatively fixed areas, we use a template rendering approach for text filling. We create background templates for these topics based on real-world images, label the fillable areas of the templates with bounding boxes, render the background templates onto the original image, and then fill the text according to the bounding box annotations of the templates.

# H. Creation of the Real Dataset.

## H.1. Data Selection Details from Existing Datasets

To ensure the quality of selected samples, we apply a rigorous filtering pipeline consisting of the following steps:

1. **Minimum Length Check:** Samples with fewer than seven words are excluded. This criterion eliminates excessively short texts that may lack meaningful content.

2. **Unique Word Ratio Check:** To promote diversity, the ratio of unique words to total words must exceed 0.3. Samples with overly repetitive word usage are filtered out.

3. **Consecutive Repetition Check:** Text containing more than three consecutive repetitions of the same word is excluded to prevent redundancy and improve coherence.

4. **Word Validity Check:** Each word must include at least one alphabetic character and be longer than one character. This ensures all words are meaningful and eliminates noise or random symbols.

5. **Text Cleaning:** Non-alphanumeric characters, except spaces, are removed. Multiple spaces are normalized into a single space to ensure the text is clean and consistently formatted.

6. **Annotation Sorting:** Annotations are ordered spatially, following a top-to-bottom and left-to-right sequence based on the coordinates of bounding polygons. This ensures spatial coherence in the text layout.

This pipeline is designed to refine the dataset and maintain high standards for text quality and diversity.

## H.2. Extracting Powerpoint Data

We extract the powerpoint data with PyMuPDF (Inc., 2025). Specifically, we transform the each page of powerpoint into pdf format, then we rephrase the powerpoint data by blocking description. For example, it split the all page into different block. Each block include elements like text or image, for text element we extract the word and for image we use the QWen-VL to generate caption and the prompt is simple *Describe this image*. For example, we simply call image.

## H.3. PPT2Details annotation generation.

In the PPT2Details subset, we use Qwen2-VL to summarize all extracted elements (text, figures, layout, etc.) into a single descriptive prompt. The text prompt for calling Qwen2-VL is:

> **Prompt Format**
>
> Given a PowerPoint slide image, extract and summarize all visual elements—such as text blocks, charts, tables, and diagrams—into a single, fluent, and logically consistent paragraph.
> You must: 1. Accurately preserve all textual content and wording details. 2. Include descriptions of all visual elements (e.g., diagrams, tables) if present. 3. Avoid omitting or paraphrasing key phrases. 4. Output only one paragraph per slide.

## H.4. Data Selection Details for TextScenesHQ Dataset

After crawling images according to topics, we use the easyOCR[1] library to recognize the text in the images. First, we save images containing more than 10 words, and then organize the text information from the upper left to the lower right to construct a JSON file. The content of the JSON file includes the text and its corresponding bounding box. During this process, some difficult data may have spelling errors, including but not limited to confusion between numbers and letters, spelling errors, and capitalization errors. In this regard, we use Llama 3.1 (Dubey et al., 2024) to check and correct the recognized text to improve the accuracy and quality of the text.

---

[1]https://github.com/JaidedAI/EasyOCR

## H.5. TextScenesHQ Image Filtering

For real image data, we primarily discarded samples where the text was not clearly visible. Additionally, samples were rejected if the detected text contained too few words (fewer than 10 in this study). We show the rejected samples in Figure 19.

## H.6. TextScenesHQ Image Annotation

After using OCR for filtering and generating bounding boxes around the text in the images, we convert the detected Chinese text and its corresponding bounding boxes into a text JSON format. Due to the diversity and complexity of the images, OCR results may contain spelling errors and misordered text. To address this, we perform three corrective steps using Llama 3.1 and Qwen 2.5-Coder. First, Llama 3.1 is used to correct any spelling mistakes in the text. Next, we use Llama 3.1 to reorder the text slightly to align with the proper syntax, as OCR typically outputs text in a left-to-right, top-to-bottom sequence without considering the multi-column layout in the images. After reordering, we generate the corrected text JSON. The third step involves addressing any potential formatting issues in the JSON. If the JSON generated in the second step is not parsable, we use Qwen 2.5-Coder to output the text JSON in markdown format to ensure proper structure.

For the image background descriptions, we use Qwen 2.5-VL to generate contextual information while preventing it from outputting any descriptions of the text within the image. Additionally, we created 500 diverse and complex scenario templates using GPT-4o to generate a wide range of image descriptions. These descriptions, combined with the corresponding text JSON, are used to generate comprehensive image information in JSON format.

## H.7. Quality classification

In this work, we mainly split the data quality according to the visual appealing semantic, and if the image include dense text and have correct captions.

# I. Annotation Details

## I.1. Joint Distribution of TextVisionBlend

Figure 20 shows the joint distribution of token numbers and image numbers in interleaved data split TextVisualBlend. We limit the number of images in a document to 4 images for clarity. The documents of TextAtlas5M contain a median number of images of 2 and a median number of tokens of 33.

## I.2. Examples from All Subsets

Our TextAtlas5M comprises a total of 10 subsets, which can be categorized into three types: *i*. Images without a specific scene, *ii*. Images with a specific scene, and *iii*. Images with a specific scene and bounding box annotations.

**Images without a Specific Scene.** For simple synthetic datasets such as Paragen-2M, where the background is plain white, we generate descriptions for image creation using prompts like: *"Please generate an image of xxx based on the following text: ."*

**Images with a Specific Scene.** This category includes images accompanied by a scene description $T$ and OCR text $O$. Using our template, we generate longer, natural descriptions by combining these elements.

**Images with a Specific Scene and Bounding Box.** For datasets like AutoSlideGen, ArxivPaper generation, and interleaved sample generation, bounding box annotations are provided for each element. In these cases, we utilize LLMs to summarize all elements into a coherent paragraph. Specifically, we include details such as bounding box coordinates and the text within each box.

All subset examples are visualized in Figure 21.

## I.3. Processing Methods

For datasets that already have captions and OCR results, such as anyword3m and mario10m, we use templates generated by GPT for concatenation (as you did before). For paragen2m, which is pure text data, we use structured sentence descriptions,

e.g., "a text white background image...". For autogen and interleave data, which are interleaved distributions, we list the text and image separately in bullet points, while placing the required elements (like bbox) and fonts in the corresponding context section. For midquality data, to ensure a natural integration, we generate scene captions using Qwen2-VL and require it to generate a render text placeholder <>, which is then replaced with the rendered text. High-quality data is processed by Llama3.1 to generate scene descriptions and optimize the OCR results (see section 3.2 for the concatenation method).

## I.4. All LDA Topics

In this section, we list top 20 LDA topics of TextAtlas5M in Table 14. Based on the topic distribution in the table, several patterns emerge:

1. **High Proportion of Common Topics**: Topics such as "Position" (15.12%), "Signs" (14.50%), and "Colors" (13.54%) account for a significant portion of the dataset. These themes likely reflect common real-world scenarios, such as signage, positioning of text and images, and the use of colors in visual communication.

2. **Content-Related Themes**: Content-centric topics like "Content" (14.79%), "Community" (8.29%), and "Safety" (2.67%) also show relatively high proportions, suggesting that the dataset includes a considerable amount of text related to information dissemination and visual design, commonly seen in advertising and informational graphics.

3. **Lower Proportion of Specialized Domains**: Topics like "Products" (1.48%), "Cloud" (0.55%), and "Shops" (0.78%) have smaller representations, indicating that the dataset covers fewer instances of text-image combinations related to specific industries or niche topics.

4. **Use of Numbers and Symbols**: Topics related to numbers, such as "Numbers" (1.75%) and "Symbols" (0.61%), occupy lower proportions, possibly reflecting that numeric and symbolic content is less prevalent in the dataset, despite its importance in some contexts.

Overall, the dataset is more focused on common visual and textual elements seen in everyday life, such as positioning, signage, and color usage, with a relatively lower emphasis on specialized topics or numeric/symbolic content.

*Table 14.* Full set of topics for the k = 20 LDA model in TextAtlas5M.

| Topic | Proportion | Keywords |
|---|---|---|
| Content | 14.79% | image, various, wall, text, several, background, includes, shows, poster, including |
| Products | 1.48% | love, product, size, case, fitness, body, water, san, bottle, products |
| Cloud | 0.55% | service, cloud, customer, things, programs, create, security, close, brooklyn, ideas |
| Food | 1.46% | coffee, food, guide, best, real, cup, home, tour, game, fresh |
| Market | 2.39% | new, x, sale, york, b, 0, f, c, car, market |
| Display | 4.88% | screen, shows, displaying, image, words, digital, options, code, display, menu |
| Travel | 2.71% | please, page, make, thank, world, one, travel, go, good, see |
| Flights | 2.84% | flight, time, gate, information, pass, numbers, details, shows, times, number |
| Map | 3.32% | map, notes, park, sticky, near, children, bus, chalkboard, road, stop |
| Books | 2.66% | board, book, display, library, books, de, reading, read, step, titled |
| Symbols | 0.61% | mounted, symbol, platform, 100, keyboard, signage, function, premium, keys, shift |
| Tickets | 4.65% | pm, ticket, train, day, date, card, weather, 12, seat, time |
| Lorem | 1.02% | lorem, ipsum, dolor, sit, amet, consectetur, ut, elit, adipiscing, sed |
| Community | 8.29% | success, school, words, community, conference, services, people, information, university, program |
| Signs | 14.50% | sign, words, picture, shows, signs, right, left, large, image, building |
| Safety | 2.67% | area, indicating, pointing, arrow, health, museum, line, parking, safety, art |
| Position | 15.12% | top, right, left, bottom, section, words, text, picture, image, icon |
| Numbers | 1.75% | 1, 2, 3, 4, 5, 6, 10, 7, 9, destination |
| Shops | 0.78% | depicts, shop, flights, counter, little, morning, synergy, scheduled, customers, eget |
| Colors | 13.54% | text, white, background, black, blue, image, red, letters, green, yellow |

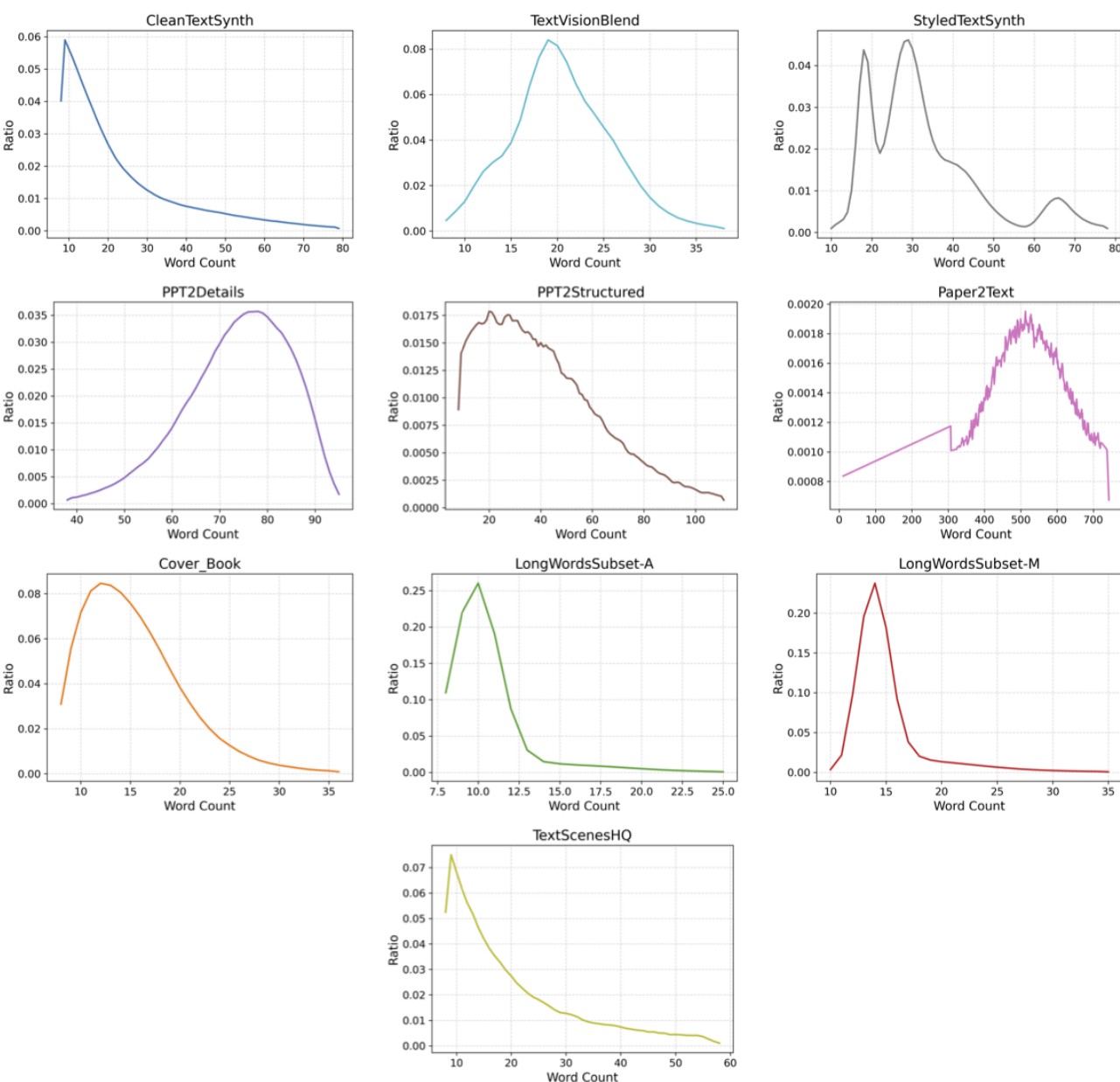

*Figure 10.* Distribution of word-count ratios across all subsets of our Dataset.

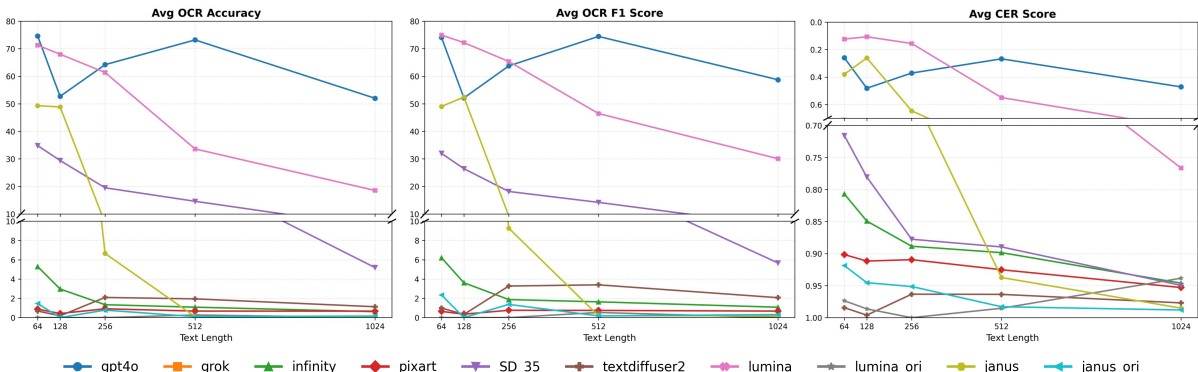

*Figure 11.* **OCR performance comparison of various models on the CleanTextSynth evaluation set across different text lengths.** We report average OCR Accuracy (left), F1 Score (middle), and Character Error Rate (CER; right, lower is better). Models *Janus* and *Lumina* represent our finetuned AR models, while *janus_ori* and *lumina_ori* refer to the original, unmodified models.

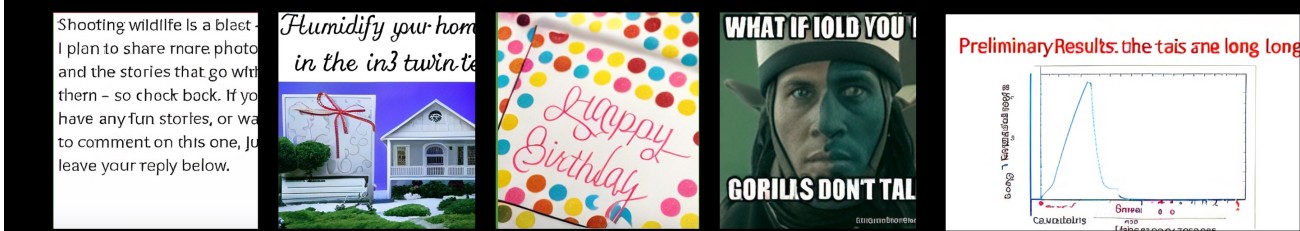

*Figure 12.* **Visualization of Lumina-mGPT (ours).** The model shows significant improvements in text rendering while maintaining strong performance on challenging tasks such as realistic face generation.

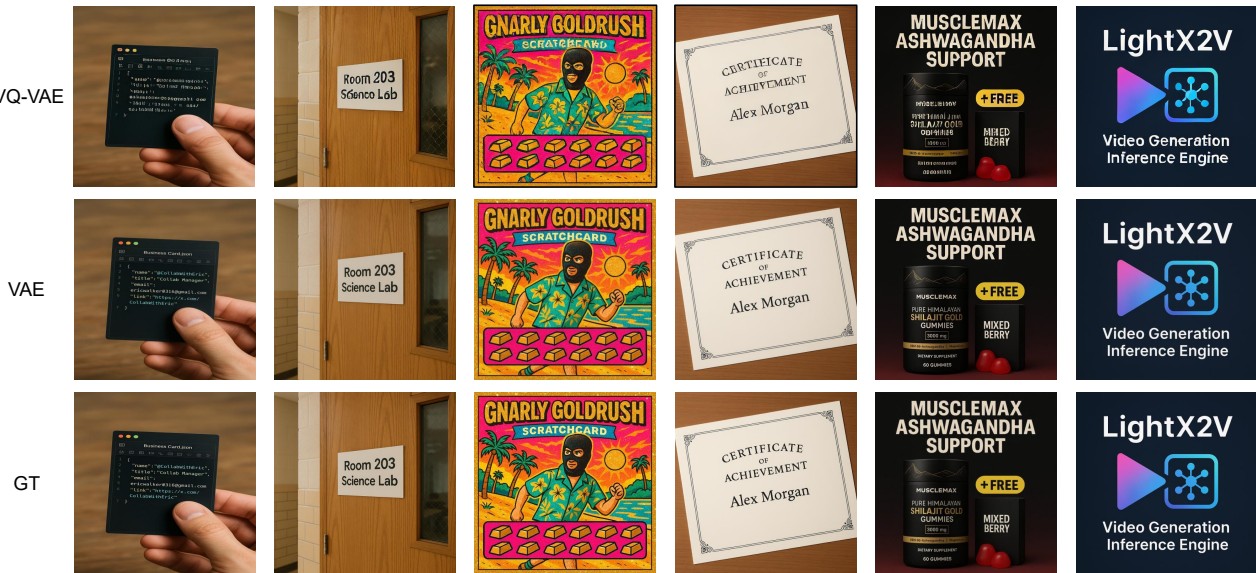

*Figure 13.* **Comparison of image reconstructions across different generative tokenizers.** Each row corresponds to a different reconstruction method: VQ-VAE from Janus Pro (Chen et al., 2025), VAE form Stable Diffusion 3.5 Large (Esser et al., 2024), and ground truth. VQ-VAE struggles with fine-grained textual detail compared to the standard VAE Method.

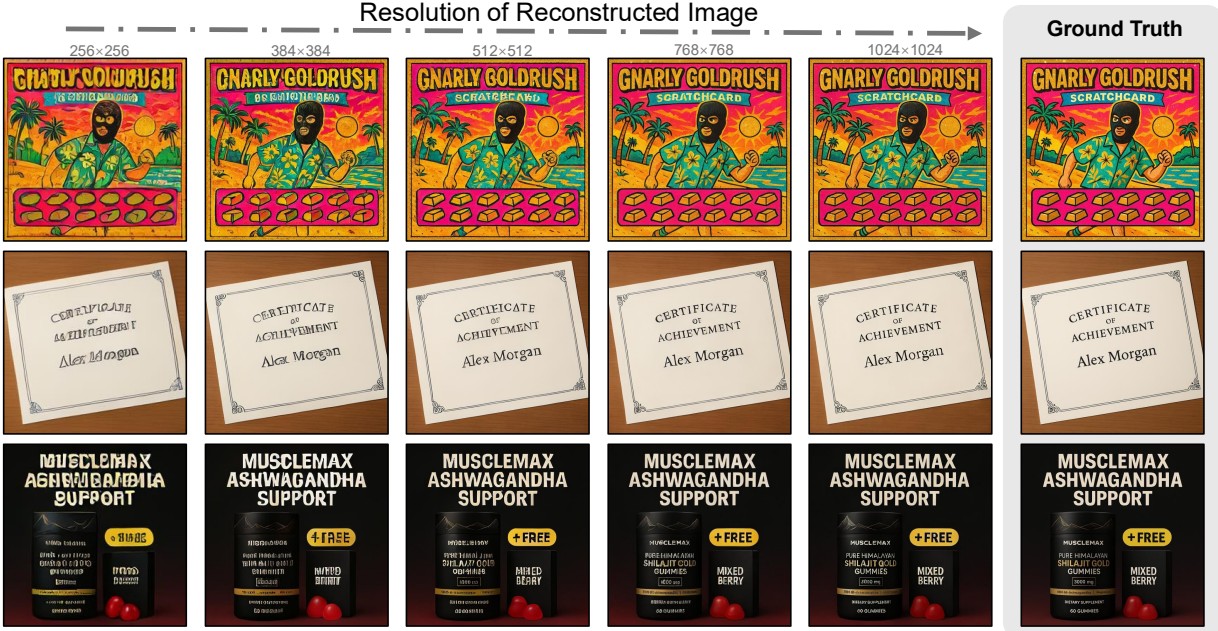

*Figure 14.* **Reconstruction results of Janus Pro's VQ-VAE tokenizer at increasing resolutions.** From left to right, image resolution increases from 256×256 to 1024×1024. Higher resolutions yield better reconstructions, especially in text clarity and layout fidelity. Ground truth (GT) images are shown in the final column for reference.

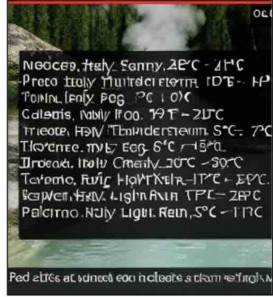

*Figure 15.* **Comparison between autoregressive and diffusion models** for text-to-image generation at $512 \times 512$ resolution. While diffusion models yield finer visual details, autoregressive models demonstrate stronger output consistency.
.

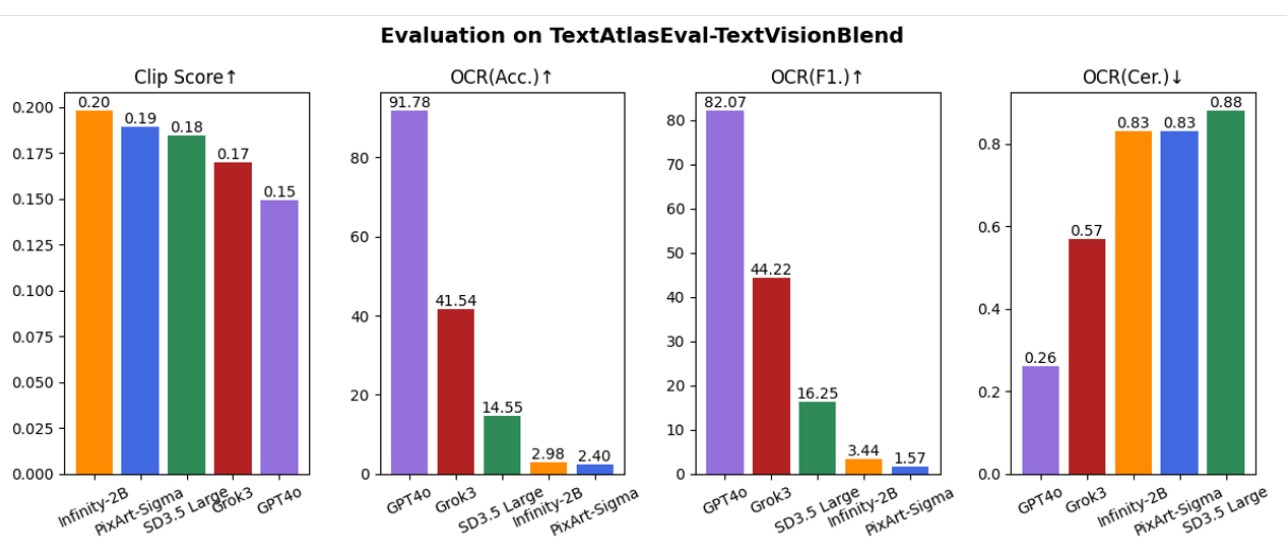

*Figure 16.* Performance comparison of state-of-the-art text-to-image generation models on our proposed benchmark, TextAtlasEval.

| Infinity-2B | Pixart | SD3.5 Large | Ground Truth |
|:---:|:---:|:---:|:---:|

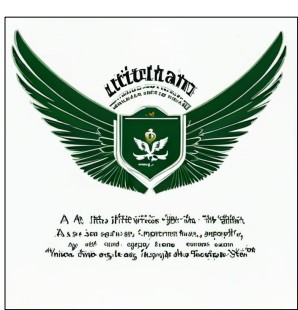
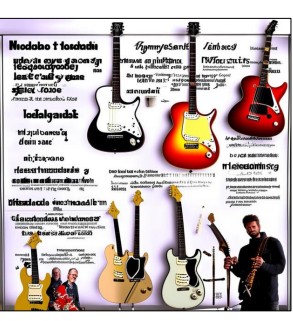
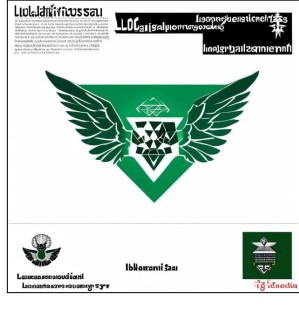
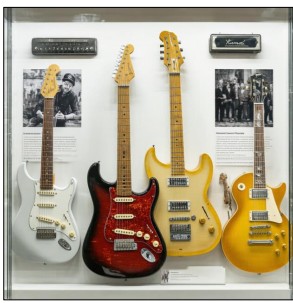
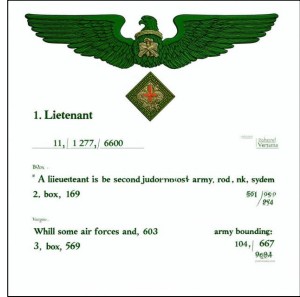
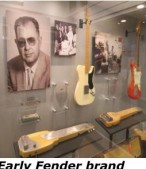
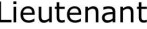
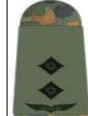

***Musical instrument***

***A musical instrument is a device created or***

*Early Fender brand electric guitars*

***The evolution of traditional musical instruments***

Lieutenant

A lieutenant is the second junior-most or in some

While some air forces use the army rank system,

*Figure 17.* **Generation example on TextVisionBlend-Eval.** SD-3.5 generates significantly more accurate text but occasionally struggles to maintain a proper interleaved layout.

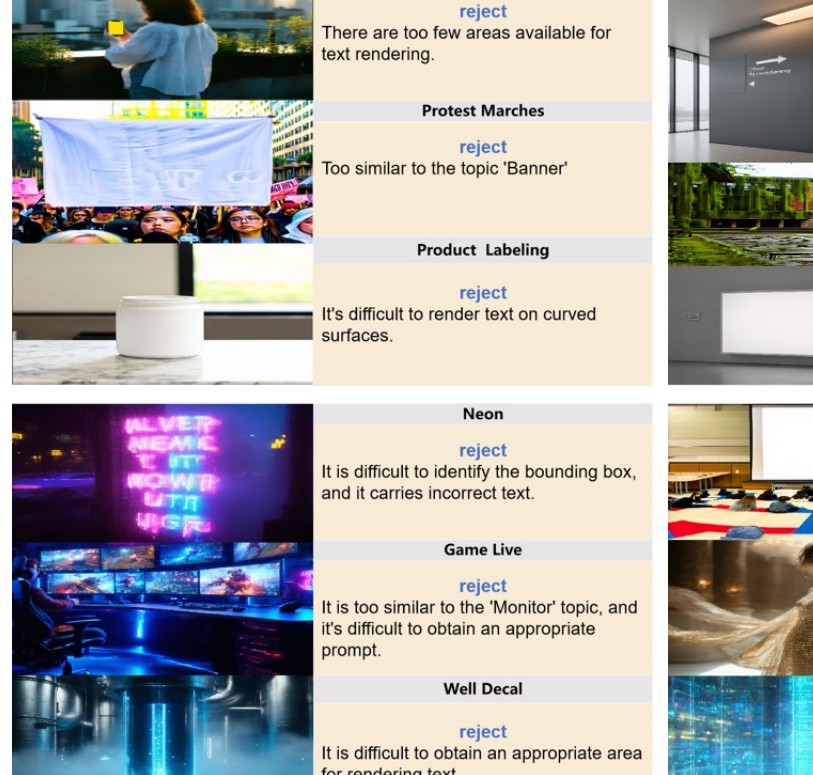
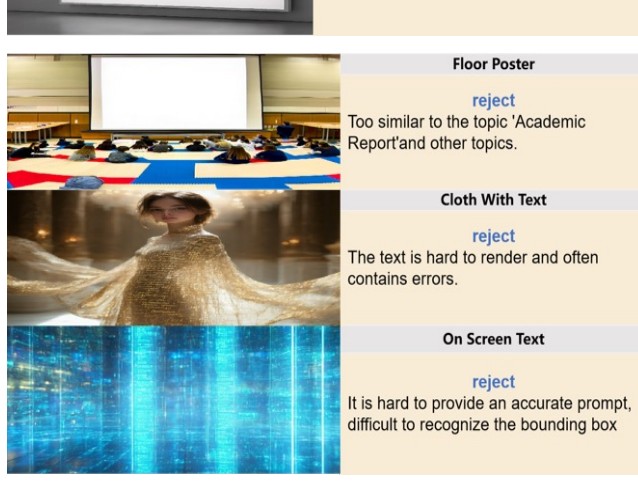

*Figure 18.* The rejected StyledTextSynth samples.

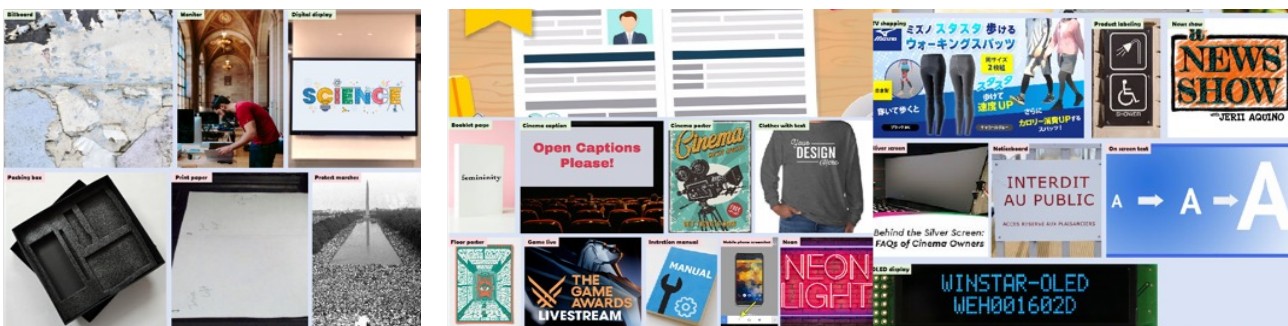

**Text Not Clearly Visible**      **Non-English Text or Fewer Than 10 Words in English**

*Figure 19.* The rejected TextScenesHQ samples.

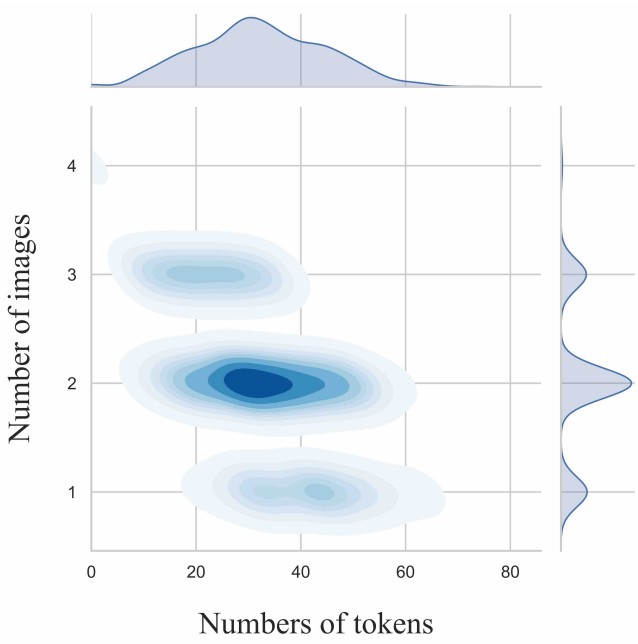

*Figure 20.* **Joint Distribution of Text Tokens and Image Count in TextVisionBlend.** Each axis is visualized alongside the corresponding marginal distributions.

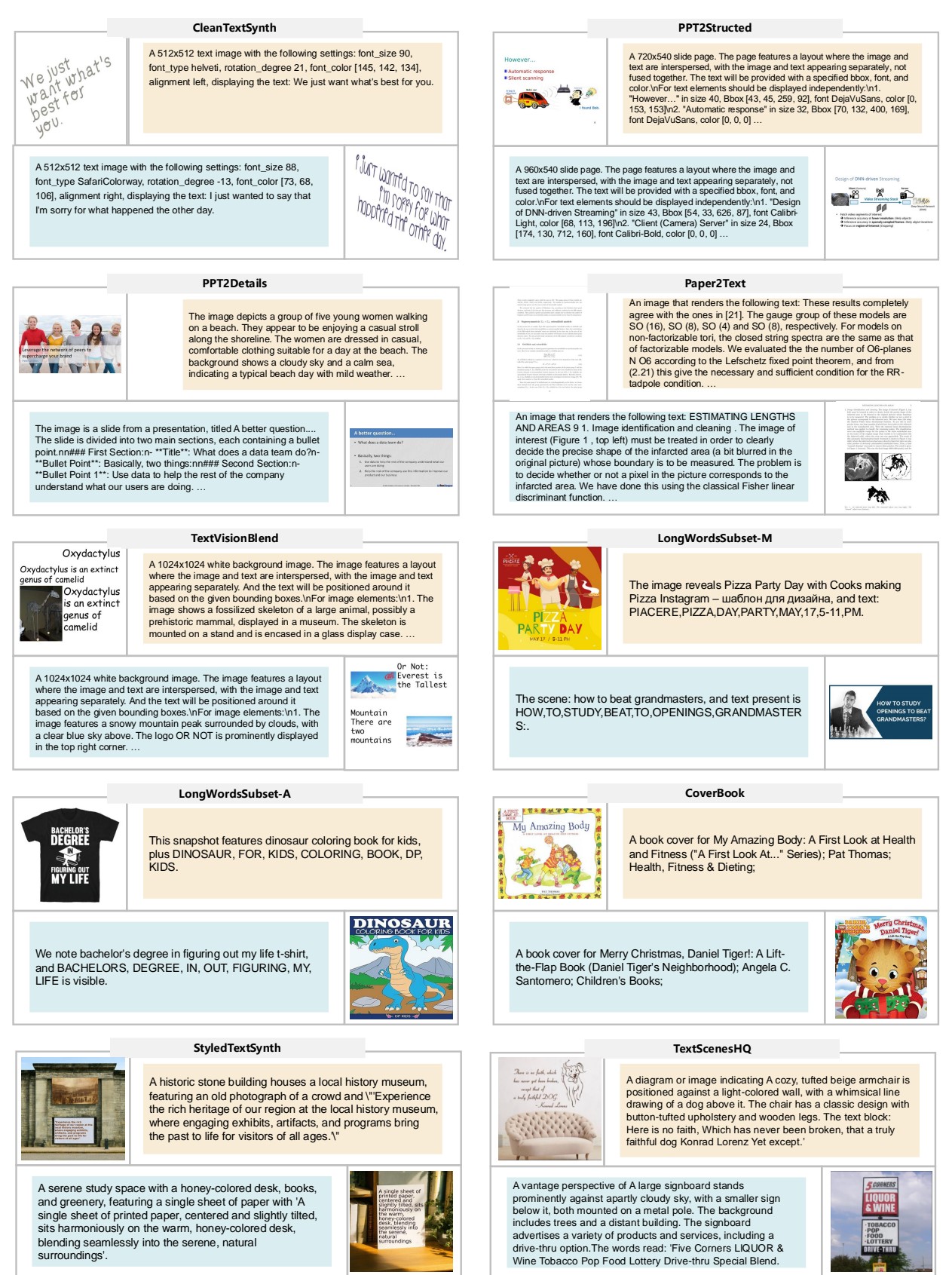

*Figure 21.* A randomly sampled selection from all subsets, including both synthetic and real data.

## J. Visualization of TextAtlas5M

### J.1. Example of StyledTextSynth Samples

To better investigate all topics included in the StyledTextSynth sample, we show the examples in Figure 22. We mainly list Blackboard Classroom, News, Banner, Sliver Screen, Notice Board, Advertisement Board, TV Shopping, Billboard, Booklet Page, Academic Report, Alumni Profiles, Tablet Screen, Printed Paper, Cinema Poster and Packing Box.

### J.2. Examples of TextScenesHQ Samples

We present examples of topics with the largest number of samples in Figure 23. These include: Product Labeling, Billboard, Packing Box, Monitor, Instruction Manual, Booklet Page, Mobile Phone Screenshot, Wall Decal, Floor Poster, Game Live, OLED Display, Protest Marches, Weather Report, Noticeboard, News Show, Blackboard Classroom, Digital Display, Cinema Caption, Wayfinding Sign, Academic Report, Alumni Profiles, Banner, Clothes with Text, and Store Sign.

These topics represent text-rich scenes commonly encountered in daily life. By applying our carefully designed filtering rules, we have ensured that only TextScenesHQ data is preserved for rendering.

## K. Annotation Quality and Error Analysis

Automatic annotations generated by large language models (LLMs) or large vision-language models (LVMs) may introduce potential noise. To mitigate this, we applied systematic quality control and calibration mechanisms across different subsets of our dataset, as detailed below.

**TextAtlasEval.** Every sample underwent manual verification, covering both the caption and downstream annotations. Samples with inaccurate or ambiguous captions were re-annotated by human annotators to ensure consistency and correctness.

**StyledTextSynth (training set).** Since the text content is directly rendered into the image, OCR annotations are guaranteed to be 100% accurate. However, bounding boxes may shift due to layout distortion. To address this, we double-checked all annotation templates and manually corrected hard cases to ensure spatial precision.

**TextScenesHQ (training set).** To avoid bias from a single model, we employed both Qwen-VL and InternVL to generate annotations. Samples with inconsistent outputs were flagged for manual review. We also performed manual template filtering to maintain high quality for short-text image pairs.

**Multi-step Calibration.** Our quality pipeline integrates human-in-the-loop correction, template validation, and cross-model consistency checks. These strategies ensure annotation reliability across key subsets including *CoverBook*, *CleanTextSynth*, *TextVisionBlend*, *StyledTextSynth*, and *TextScenesHQ*. We will further include annotation quality statistics in the Appendix to provide transparency.

**PPT2Details, PPT2Structured, and Paper2Text.** For these subsets, annotations are extracted from original PDF files using PyMuPDF. Since structural information such as element positions is derived directly from the source, these annotations are highly accurate. LLMs are only used to generate image captions and summary sentences, where they are particularly well-suited, thus introducing minimal error.

**LongWordsSubset.** This subset does not rely on human annotations or LLMs. Instead, it is constructed by filtering existing datasets based on length and layout heuristics, ensuring high quality at scale with minimal noise.

In summary, the combination of manual verification, template validation, and cross-model consistency checks provides robust safeguards against annotation errors. These procedures ensure that our dataset maintains high annotation quality across both training and evaluation splits.

## L. LLM Usage Statement

Large language models (LLMs) were used in this work under two restricted scenarios:

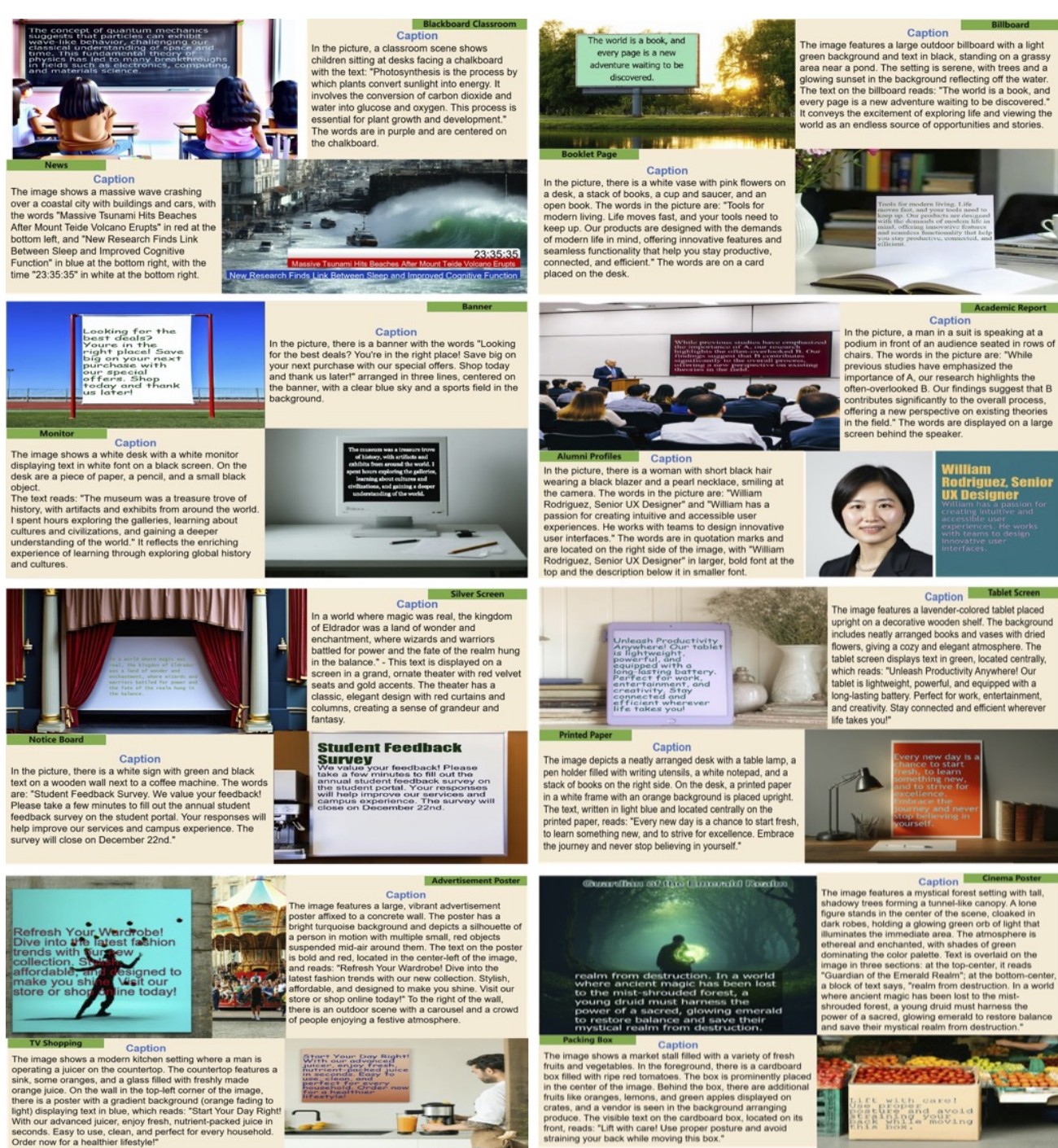

*Figure 22.* StyledTextSynth examples.

**Product labeling**

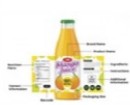

The image shows a bottle of "Mango Juice" with a label template on the left side, featuring sections for "Nutrition Facts," "Contact Information," and a "Barcode." The label includes "Brand Name," "Product Name," "Ingredients," "Instructions," "Additional Information," and "Packaging Size." The words are arranged as follows: "Mango Juice" is at the center, "100% NATURAL" is below it, "NO SUGAR ADDED" is on the right side, and "LOL" is above the brand name.

**Packing box**

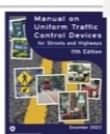

The picture shows a pink box for a "FANTECH SAKURA EDITION GAMING SET" with the hashtag "#GEARUPANDWIN" and logos for "FANTECH" and "FANTECH WORLD" on the front and back. The words are located on the left side of the box and the front of the box.

**Instruction manual**

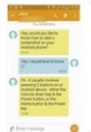

The picture shows a cover of the "Manual on Uniform Traffic Control Devices for Streets and Highways 11th Edition" by the U.S. Department of Transportation Federal Highway Administration, dated December 2023. The words are in the center of the cover, with the title in large white letters on a dark blue background, and the edition and date below it. Surrounding the title are images of various traffic signs and street scenes.

**Mobile phone screenshot**

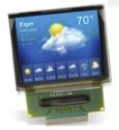

The picture shows a conversation on a messaging app with Jane, dated Thursday, April 28, 2016. The words are: "Hey, would you like to know how to take a screenshot on your Android phone?" (yellow text, 13:18), "Yes, I would love to know :)!" (blue text, 13:19), and "Ok - it usually involves pressing 2 buttons on yr Android device - either the Volume down key & the Power button, or the Home button & the Power key" (yellow text, 13:20).

**Floor poster**

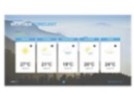

A promotional sign on a wooden floor with the words "ON THE AVENUE SPECIAL OFFERS 25% OFF SEASONAL SALE" in the center, and "976.555.6588" and "WWW.ONTHEAVENUE.WEB" at the bottom.

**OLED display**

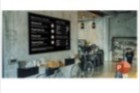

The picture shows a small electronic device with a screen displaying a weather forecast. The screen shows a sunny sky with clouds and the text "Elgin Saturday 70°F" at the top left, with icons indicating humidity, wind speed, and pressure below. The days of the week are shown at the bottom with corresponding weather icons. The words are: "Elgin Saturday 70°F" at the top left, "SUN MON TUES WED THUR FRI SAT" at the bottom.

**Weather report**

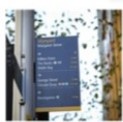

The image shows a weather forecast for Lagos with icons and temperatures for Monday through Friday, set against a mountainous background. The words are: "WEATHER FORECAST" at the top, "Weather" and "LAGOS" below it, and "MONDAY" through "FRIDAY" above each day's forecast. The temperatures are "27°C" for Monday, "21°C" for Tuesday, "19°C" for Wednesday, "20°C" for Thursday, and "24°C" for Friday. The website "WWW.POWERLIDES.COM" is at the bottom.

**News show**

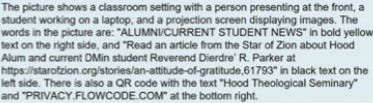

The picture features a blue background with a world map and the words "JOIN US FOR THE UPCOMING GRADUATE INFORMATION SESSION" in white text on the left side. On the right side, there is a white circle with the words "VIRTUAL - ZOOM" in red text at the top, followed by "21 Sep 2024 (Saturday)" in black text, and "10:30 AM - 12:00 PM (GMT+8)" in black text below. At the bottom of the circle, there is a red button with the text "Register at https://bit.ly/WKWSCIInfoSession24" in white text.

**Digital display**

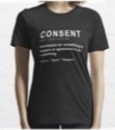

The picture shows a café interior with a menu board on the wall, two bicycles, and shelves with various items. The words in the picture are: "Specialty Drinks," "Cherries of Fire," "Pumpkin Spice Latte," "S'mores Latte," "Peppermint Mocha," "Daily Brews," "Hot Coffee," "Cold Brew," "Mocha," "Espresso Drinks," "Macchiato," "Flat White," "Cappuccino," "dite," and "P." The words are arranged in two columns on the menu board, with "Specialty Drinks" and "Daily Brews" at the top, followed by the drink names and prices. The word "P" is in a red circle in the bottom right corner.

**Wayfinding sign**

The picture shows a blue street sign with directions and distances to various locations, including "Wynyard Margaret Street," "Millers Point," "The Rocks," "Walsh Bay," "George Street," "Circular Quay," and "Barangaroo," with icons indicating walking and public transport options. The words are arranged in a list format with arrows and symbols indicating directions and travel times.

**Alumni Profiles**

The picture shows a classroom setting with a person presenting at the front, a student working on a laptop, and a projection screen displaying images. The words in the picture are: "ALUMNI/CURRENT STUDENT NEWS" in bold yellow text on the right side, and "Read an article from the Star of Zion about Hood Alum and current DMin student Reverend Dierdre' R. Parker at https://starofzion.org/stories/an-attitude-of-gratitude,61793" in black text on the left side. There is also a QR code with the text "Hood Theological Seminary" and "PRIVACY.FLOWCODE.COM" at the bottom right.

**Clothes with text**

A person is wearing a black T-shirt with the words "CONSENT noun. [ken sent] Permission for something to happen or agreement to do something." and "See also: 'Teach', 'Respect'" in white text, with "CONSENT" at the top, "Permission for something to happen or agreement to do something." in the middle, and "See also: 'Teach', 'Respect" at the bottom.

**Billboard**

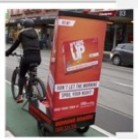

A person on a bicycle is towing a red cart with the words "HEADS UP" and "DON'T LET THE MORNING SPOIL YOUR NIGHT!" on it, along with "FOR THE SYMPTOMATIC RELIEF OF HANGOVER" and "GRAB YOURS TODAY AT compounding PHARMACY" and "RUNNING BOARDS" with a phone number "1300 334 556" on the side. The words are on the side of the cart.

**Monitor**

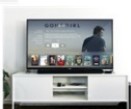

A modern living room with a white TV stand, a large flat-screen TV displaying the movie "Gone Girl" with the text "GONE GIRL" at the top, a potted plant on the left, and a colorful framed artwork on the wall. The words are: "GONE GIRL" at the top of the TV screen, "DIRECTOR David Fincher" and "STARRING Ben Affleck, Rosamund Pike, Neil Patrick Harris, Kim Dickens" below the movie title, and "Available On iTunes HBO" at the bottom.

**Booklet page**

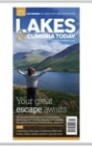

A person with arms outstretched stands on a cliff overlooking a lake and mountains, with the words "LAKES & CUMBRIA TODAY" at the top, "Your great escape awaits" in the center, and "ONLY £3.50" and "220 PAGES OF DAYS OUT AND ACTIVITIES" at the top left, "SPRING/SUMMER 2022" at the top right, and "PLUS" and "Celebrating 1900 years of Hadrian's Wall with events, history, family activities, and lots more" at the bottom left.

**Wall Decal**

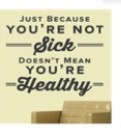

The picture features a beige background with a quote in black text that reads, "Just because you're not sick doesn't mean you're healthy." The words "sick" and "healthy" are emphasized in a cursive style, while the rest of the text is in a sans-serif font. The quote is centered in the image, with "sick" and "healthy" placed at the center of the text block.

**Game live**

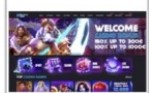

The image features a promotional banner for "MyStake" with a "WELCOME CASINO BONUS" offer of "150% UP TO 300C" and "100% UP TO 1000C" in large, bold text on the right side. The words are in quotation marks. The top menu includes options like "Sports," "Live Sports," "Casino," "Live Casino," "Virtual," "Racing," "E-Sports," "Tournaments," "Originals," "Mini Games," and "Promotions." Below, there are icons labeled "SPORT," "CASINO," "MINI GAMES," and "LIVE CASINO," with a "VIEW ALL" button on the right. The bottom section showcases "TOP CASINO GAMES" with various game icons.

**Protest marches**

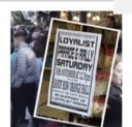

The picture shows a crowded outdoor market with people walking around, some wearing winter clothing and carrying bags. In the foreground, there is a sign with the words "LOYALIST PARADE & RALLY SATURDAY 30th NOVEMBER AT 12.30pm FROM SANDY ROW ORANGE HALL RALLY AT CITY HALL IN SUPPORT OF LOYALIST AREAS 125TH ANNIVERSARY OF THE FLAG PROTEST" in quotation marks. The sign is located in the upper right corner of the picture.

**Noticeboard**

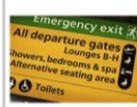

The picture shows a yellow sign with the words "Emergency exit" at the top in green, followed by "All departure gates" and "Lounges B-H" in black text, and "showers, bedrooms & spa" and "Alternative seating area" in smaller black text. Below, there are symbols for toilets and a wheelchair, with the word "Toilets" in black text.

**Blackboard classroom**

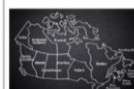

A chalk-drawn map of Canada with the words "Yukon," "Northwest Territories," "Nunavut," "British Columbia," "Alberta," "Saskatchewan," "Manitoba," "Ontario," "Quebec," "Newfoundland," "Prince Edward Island," "Nova Scotia," and "New Brunswick" labeled in white chalk, with "Yukon" in the top left, "Northwest Territories" and "Nunavut" to its right, "British Columbia" and "Alberta" in the bottom left, "Saskatchewan" and "Manitoba" in the center, "Ontario" and "Quebec" in the middle right, "Newfoundland" in the top right, "Prince Edward Island" and "Nova Scotia" in the bottom right, and "New Brunswick" below "Nova Scotia."

**Cinema caption**

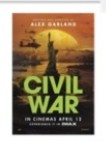

The image features the Statue of Liberty with helicopters flying around it against a dramatic orange sky, with the text "WRITTEN AND DIRECTED BY ALEX GARLAND" at the top, "CIVIL WAR" in large green letters in the center, and "IN CINEMAS APRIL 12 EXPERIENCE IT IN IMAX" at the bottom.

**Academic report**

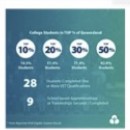

The image shows a group of students with the text "College Students in TOP % of Queensland" at the top. Below are four circular icons with percentages: "TOP 10%" (14.3% Students), "TOP 20%" (51.4% Students), "TOP 30%" (71.4% Students), and "TOP 50%" (82.9% Students). Below these, it states "28 Students Completed One or More VET Qualifications" and "9 School based Apprenticeships or Traineeships Secured / Completed." The bottom notes "* From Reported ATAR Eligible Student Results."

**Banner**

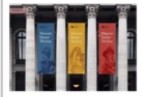

The picture shows three banners hanging between columns in front of a building, with people gathered below. The words in the picture are: "FREE" in red text with a downward arrow, and "Museum Banner Mockup" in white text on each banner. The "FREE" text is in the top left corner, and "Museum Banner Mockup" is on each banner.

**Store sign**

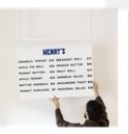

A person is holding a menu titled "HENRY'S" at the top, with a list of food items and prices: "GRANOLA YOGURT $16," "APPLE PIE ROLL $18," "PEANUT BUTTER $16," "APPLE YOGURT $18," "BUTTER GRANOLA $22," "PEANUT PORRIDGE $9," "BREAKFAST ROLL $15," "SMOKED BUTTER $20," "FRUIT ROLL $15," "GRANOLA SALAD $22," "MUSHROOM TOAST $20," "SEASONAL SALAD $18."

*Figure 23.* TextScenesHQ topics.

**Manuscript Preparation.** LLMs were employed solely as auxiliary tools for surface-level editing, including grammar correction, minor rephrasing, and stylistic refinement to improve readability. They were never used to generate research ideas, methodologies, or conclusions.

**Dataset Construction.** For subset TextScenesHQ of our dataset, we used GPT-4o to generate a small collection of seed topics that guided subsequent data curation. In addition, multimodal models such as Qwen2-VL were employed to produce descriptive annotations for image-containing subsets (e.g., *PPT2Details*), where visual and textual elements had to be summarized into coherent descriptions. These model-generated annotations were further calibrated with template rules and human verification to ensure quality and consistency.

**Benchmark Evaluation.** To assess the reliability of our proposed benchmark, we used state-of-the-art models such as GPT-4o, Grok3, and Qwen2-VL as evaluation agents. These systems provided OCR-based recognition results and performance baselines against which open-source models were compared. This usage was strictly limited to evaluation purposes and does not affect the integrity of dataset construction.

Beyond the above cases, no other use of LLMs was involved in data generation, experimental design, or analysis.

