# OpenReview forum: "TextAtlas5M: A Large-Scale Dataset for Long Text Image Generation"
_ICML.cc/2026/Conference — ICML 2026 regular_

### Official Review · Reviewer_dbxy · 2026-03-07

**Soundness:** 3
**Presentation:** 3
**Significance:** 3
**Originality:** 3
**Overall Recommendation:** 4
**Confidence:** 4

**Summary:**

An important context analyzed by the paper is that modern text-to-image models struggle with long and dense text rendering due to the lack of high-quality dedicated datasets. It then contributes TextAtlas5M, a 5M-scale dataset with diverse real and synthetic images, along with the human-refined TextAtlasEval benchmark. Extensive experiments validate that fine-tuning on TextAtlas5M significantly improves text rendering for both autoregressive and diffusion models. Overall, the research considers an important area in text-rich image generation and offers a valuable resource to push forward related studies.

**Compliance With Llm Reviewing Policy:**

Affirmed.

**Final Justification:**

My concerns have been solved and I will maintain my rating.

**Key Questions For Authors:**

see weaknesses

**Strengths And Weaknesses:**

**Strengths**

1. The paper uses a systematic pipeline including automatic filtering, LLM/VLM annotation, and human verification to ensure data quality. Experiments cover comprehensive metrics (OCR accuracy, CER, CLIP score, layout matching) and strong baselines, with ablation studies validating the effectiveness of mixed real-synthetic training; all claims are well-supported by quantitative results.

2. The work redefines “long text” beyond length to include layout complexity and semantic richness. It innovatively combines synthetic clean text, interleaved text-image data, and real-world dense-text scenes, creating a diverse dataset structure unseen in prior short-text-focused resources.

3. The paper is well-organized with logical flow from problem motivation to data construction, evaluation, and analysis. Abundant figures, tables, and examples visualize dataset composition and model performance; key pipelines (data filtering, annotation, rendering) are detailed enough for expert reproduction.

**Weaknesses**

1. Although human verification is applied to the test set, the paper does not provide a quantitative analysis of annotation error rates or noise levels in the large-scale training subsets. Potential errors from OCR and LLM/VLM labeling are not rigorously measured, leaving uncertainty about data purity.

2. The core contribution is a dataset rather than a new algorithm. The data construction pipeline combines existing techniques (OCR, LLM annotation, layout rendering, filtering) without introducing novel theoretical frameworks or new generative mechanisms, leading to incremental methodological novelty.

---

> ### Author Rebuttal · Authors · 2026-03-31
>
> ## Response to Reviewer dbxy
> We sincerely thank Reviewer dbxy for the positive assessment and constructive feedback. We address each point below.
>
> ### **W1. Error rates in training subset**
> We randomly sampled 100 images from each subset and manually verified annotation correctness (OCR text and VLM captions). The results show consistently low error rates across subsets, with fewer than 4 errors per subset(>96% accuracy), indicating that **annotation noise in the training data is limited and well-controlled**.
>
> ### **W2. Methodological novelty**
> We thank the reviewer for the comment regarding methodological novelty. We agree that our work is primarily data-centric, and would like to clarify its contribution in this context.
>
> In this work, we
> - identify dense long-text image generation as a distinct and underexplored setting, supported by the large gap in token length between existing datasets (e.g., 9.92 in AnyWords3M and 16.13 in Mario 10M vs. 148.82 in TextAtlas5M),
> - design a structured multi-stage data pipeline (including SAM2-guided region detection and perspective-aware rendering) to generate diverse and controllable training data beyond simple collection,
> - introduce TextAtlasEval and a layout-aware metric to better capture structural consistency beyond OCR-based evaluation.
>
> Also, our experiments show that these design choices lead to substantial and consistent improvements across multiple model families and external benchmarks. For example, Pixart-α improves Text Score on LongText-Bench from 0.14 to 24.28, and Word Accuracy on CVTG-2K from 0.02 to 31.30 (Table 4).
>
> We believe that establishing the correct data and evaluation paradigm is a necessary and impactful step toward future algorithmic advances in this area.

---

> > ### Author Rebuttal · Reviewer_dbxy · 2026-04-03
> >
> > Overall Recommendation:

---

### Official Review · Reviewer_iJtQ · 2026-03-10

**Soundness:** 3
**Presentation:** 2
**Significance:** 3
**Originality:** 3
**Overall Recommendation:** 4
**Confidence:** 4

**Summary:**

This paper introduces **TextAtlas5M**, a 5M-scale dataset for dense and long-text image generation, along with **TextAtlasEval**, a 4,000-sample human-refined benchmark. The paper is motivated by the limitation of existing text-rich image datasets, which mainly focus on short and simple text, and shows that current generation models still struggle with long text rendering, complex layouts, and text-image alignment. The authors further benchmark multiple open- and closed-source models and show that training on TextAtlas5M improves both diffusion and autoregressive models on dense-text generation tasks. Overall, the research considers an important area, and an important context analyzed by the paper is the gap between existing short-text benchmarks and real-world text-rich scenarios such as documents, slides, and posters.

**Compliance With Llm Reviewing Policy:**

Affirmed.

**Final Justification:**

Thank you for the detailed rebuttal. My concerns have been adequately addressed, especially regarding the dataset construction details, layout-related evaluation, and the additional evidence on stronger models and comparisons with other datasets. Overall, I find the rebuttal convincing and believe the proposed dataset and benchmark can be valuable resources for future research on text-rich image generation. I therefore update my score to 4 (Weak Accept).

**Key Questions For Authors:**

See the weakness part.

**Limitations:**

See the weakness part.

**Strengths And Weaknesses:**

### Strength
1. The paper studies **long-text rendering in image generation**, which is a promising and relatively underexplored problem.
2. The proposed dataset is **large-scale**, which could serve as a useful resource for training and evaluating models on text-rich image generation.
3. The paper introduces a **dedicated benchmark (TextAtlasEval)** with human-refined annotations, which helps provide a more reliable evaluation of dense-text image generation.

### Weakness
1. The discussion of text-to-image generation in the related work section is incomplete. For a paper submitted in early 2026, the related work only covers image generation models up to 2024. Many representative models released in 2025 are missing, such as the GPT-Image series, Qwen-Image series, Nano-Banana series, and FLUX.2 series, as well as several unified multimodal models including BAGEL, HunyuanImage-3.0, the Show-o series, and the Janus series. I also notice that some of these 2025 models (e.g., GPT-Image, Nano-Banana, Qwen-Image) are included in the experimental evaluation, which further suggests that they should also be discussed in the related work section.
1. The paper categorizes image generation models into diffusion models and autoregressive (AR) models. However, this categorization seems somewhat outdated given the current landscape. In the paper, the so-called unified models (e.g., Chameleon and Janus) are treated as AR models, but unified models are not necessarily purely autoregressive. For example, models such as Show-o, BAGEL, and Transfusion adopt hybrid architectures. Therefore, the current categorization may oversimplify the architectural diversity of modern image generation systems.
1. The comparison presented in Figure 1 is not sufficiently clear. It is difficult to identify which images correspond to the baseline dataset and which correspond to TextAtlas5M. The figure would benefit from clearer labeling or visual cues.
1. The dataset construction process relies heavily on Qwen2-VL for annotation. Although the authors do not explicitly specify the model size, it appears likely that Qwen2-VL-7B is used (as mentioned in Sec. G.5.1). While this choice may have been reasonable when the dataset was initially constructed, there are concerns regarding the accuracy of the generated captions. Compared with more recent open-source VLMs such as Qwen3.5 or Qwen3 (which had already been open-sourced for several months before submission), Qwen2-VL-7B differs significantly in terms of model scale, version, and reasoning capabilities. This raises questions about the reliability of the annotations produced by this model.
1. In lines 130–131, the paper states: “We then identify suitable text placement areas, refining them with human annotations as needed.” However, it is unclear how these “suitable text placement areas” are automatically identified before human refinement. More details about this process would improve the clarity and reproducibility of the dataset construction pipeline.
1. Based on my understanding, both the TextAtlas5M training set and the TextAtlasEval benchmark mainly focus on the accuracy of long-text rendering. However, in many real-world scenarios (e.g., posters, menus, or product images), generating multiple text regions within a single image is also important. In these cases, text layout and spatial arrangement are critical in addition to text content accuracy. Recent benchmarks such as OneIG-Bench [1] and T2I-CoReBench [2] emphasize these aspects. It would be helpful to clarify whether the proposed dataset considers such complex text layout scenarios.
1. In the experiments, the authors mainly fine-tune three models: Janus-Pro-1B, Lumina-mGPT-7B, and PixArt-α-0.6B, and report significant improvements after SFT. However, these models are not known for strong text-rendering capabilities in their pretrained form, likely due to limited exposure to text-rich training data. Therefore, the observed improvements may not be particularly surprising, as even relatively simple text-rendering datasets could potentially yield similar gains. To better demonstrate the advantages of the proposed dataset, it would be more convincing to perform SFT on models that already have relatively strong text-rendering capabilities, such as Qwen-Image, Z-Image, or FLUX.2-klein.



[1] Oneig-bench: Omni-dimensional nuanced evaluation for image generation, NeurIPS'25

[2] Easier Painting Than Thinking: Can Text-to-Image Models Set the Stage, but Not Direct the Play?, ICLR'26

---

> ### Author Rebuttal · Authors · 2026-03-31
>
> ## Response to Reviewer iJtQ
>
> ###  **W1. Related work.**
> We thank the reviewer for this helpful suggestion. We will include more recent image generation models in the related work section and incorporate this update in the revised version of the paper.
>
> ### **W2. Category image generation model.**
> We thank the reviewer for this helpful suggestion. We agree that the diffusion vs. autoregressive categorization is simplified and does not fully capture hybrid architectures such as Show-o, BAGEL, and Transfusion. We will revise the paper accordingly.
>
> ### **W3. Fig1 Refinement.**
> We thank the reviewer for this suggestion. We will refine the Fig.1 for better understanding in the revised version.
>
> ### **W4. Annotation reliability**
> We thank the reviewer for raising the concern about annotation quality. We would like to clarify that Qwen2-VL is primarily used for generating descriptive captions in our pipeline, rather than performing complex reasoning.
>
> Furthermore, our dataset construction does not rely solely on a single model. Instead, we employ different quality control processes on each subset to ensure annotation reliability. Importantly, models trained on TextAtlas5M achieve consistent improvements across different architectures and external benchmarks, which would be difficult to achieve if the annotation were unreliable. This provides indirect evidence of the overall reliability of the dataset.
>
> ### **W5. Text placement area**
>
> We thank the reviewer for pointing out this ambiguity and apologize for the lack of clarity in the main text. The automatic identification of text placement areas is primarily implemented using YOLOv11 and SAM2, and the detailed implementation is described in Appendix G.4.
>
> ### **W6. Layout consideration**
>
> We thank the reviewer for highlighting the importance of multi-region layout and spatial arrangement. We would like to clarify that our dataset explicitly considers complex layout scenarios during construction, rather than focusing solely on long-text accuracy.
>
> * **Training:** During training data construction: We deliberately include subsets such as PPT2Details and CoverBook that naturally exhibit complex layouts. In addition, TextVisionBlend is designed to generate interleaved text-image compositions with multiple text regions (with **an average of **3.25** text regions per image**). Furthermore, in TextScenesHQ, we curate topics that typically involve complex layouts (as shown in Fig. 5).
> * **Evaluation:** During evaluation: We introduce a layout-aware metric (IOU Count) to measure region-level consistency. As shown in Table 3, models trained on TextAtlas5M achieve significant improvements in layout matching (e.g., Lumina-mgpt improves from 29.5% to 84.3% on TextVisionBlend), demonstrating the effectiveness of our dataset in improving layout generation.
>
> We agree that more fine-grained layout evaluation is an important direction and will explore this in future work.
>
> ### **W7. Exposure training data**
>
> We thank the reviewer for this constructive suggestion.
>
> We respectfully note that the reviewer's concern is based on the assumption that "any simple text-rendering dataset would yield similar gains." However, this is directly contradicted by our existing and additional experiments.
>
> In Table 6, PixArt-α fine-tuned on TextAtlas5M consistently outperforms the same model fine-tuned on Mario10M on LAIONEval4000, despite Mario10M being larger (10M vs. 5M samples). Specifically, TextAtlas5M achieves substantially higher performance (e.g., Acc: 67.53 vs. 52.37), demonstrating that the **improvement is driven by data design rather than data scale alone**.
>
> To further support this, we provide a direct comparison on TextAtlasEval:
>
> | Data | CleanTextSynth Acc | StyleTextSynth Acc |
> |--------------|-------------------|-----------------|
> |  Pixart-α| 0.23 | 0.05 |
> | + Mario 10M | 3.72 | 1.24 |
> | + TextAtlas5M | **39.14** | **22.10** |
>
> TextAtlas5M substantially outperforms Mario10M, with particularly large gaps on long-text scenarios. This clearly demonstrates that the **gains arise from the specific design of TextAtlas5M, rather than simply the presence of additional text data**.
>
> **Additional experiments (stronger model)**
> To further address the reviewer's concern, we conduct additional experiments on a stronger model, FLUX.2-klein-4B. Due to computational constraints, we adopt a LoRA-based fine-tuning strategy and report results on the challenging StyledTextSynth subset.
>
> | Model                     | Acc. |
> |-|-|
> | Flux2 klein 4B|37.59|
> | Flux2 klein 4B(finetuned) | **40.12**|
>
> We observe a clear improvement in accuracy. This improvement is achieved on a relatively strong base model, suggesting that the benefit of TextAtlas5M is not limited to weak models with poor text exposure.

---

> > ### Author Rebuttal · Reviewer_iJtQ · 2026-04-03
> >
> > Thank you for the detailed rebuttal. My concerns have been adequately addressed, especially regarding the dataset construction details, layout-related evaluation, and the additional evidence on stronger models and comparisons with other datasets. Overall, I find the rebuttal convincing and believe the proposed dataset and benchmark can be valuable resources for future research on text-rich image generation. I therefore update my score to 4 (Weak Accept).

---

### Official Review · Reviewer_fs8n · 2026-03-12

**Soundness:** 3
**Presentation:** 3
**Significance:** 3
**Originality:** 2
**Overall Recommendation:** 4
**Confidence:** 3

**Summary:**

This paper introduces TextAtlas5M, a large-scale dataset designed to improve image generation models that need to render long and complex text. The authors observe that current text-to-image models struggle with images containing dense or structured text because existing datasets mainly contain short and simple text. To address this gap, they build a dataset of 5 million images that includes both synthetic and real-world text-rich images, covering scenarios such as documents, posters, slides, and other layouts where text and visual elements are interleaved. The dataset also includes a curated evaluation benchmark called TextAtlasEval with about 4,000 challenging examples to test how well models generate long text and complex layouts. Experiments show that many state-of-the-art models still perform poorly on this benchmark, but fine-tuning models on TextAtlas5M significantly improves their ability to generate readable text and handle complex layouts, demonstrating the value of the dataset for advancing text-rich image generation.

**Compliance With Llm Reviewing Policy:**

Affirmed.

**Final Justification:**

My concerns have been addressed, I will maintain my score.

**Key Questions For Authors:**

1. What mechanisms were used to ensure the quality and correctness of the rendered text?
2. Could the authors compare models fine-tuned on TextAtlas5M with the same models fine-tuned on other existing text-image datasets? Such a comparison would help demonstrate whether the improvements come from the specific design of TextAtlas5M (e.g., long text and complex layouts) rather than simply from additional training data.

**Limitations:**

yes

**Strengths And Weaknesses:**

## Strengths

- The paper addresses an important limitation in current text-to-image generation systems.
- The proposed dataset is large-scale (5M images) and specifically designed for dense-text image generation, filling a gap left by prior datasets that mainly contain short and simple text.
- The dataset includes both real-world and synthetic images with diverse layouts (documents, posters, slides, etc.), which improves coverage of practical text-rich scenarios.
- The authors also introduce a dedicated evaluation benchmark (TextAtlasEval) with carefully curated challenging cases, enabling more rigorous evaluation of text rendering capabilities.
- Experimental results show that fine-tuning models on TextAtlas5M significantly improves performance on multiple benchmarks, demonstrating the practical usefulness of the dataset.
- The dataset appears useful for both diffusion-based and autoregressive image generation models, making it broadly applicable.

## Weaknesses

- The contribution mainly lies in dataset construction rather than proposing new algorithms or modeling techniques, which limits the methodological novelty.
- The dataset may still have limited coverage for certain specialized domains, such as scientific documents, multilingual text, or formula-heavy layouts.
- The evaluation primarily focuses on improvements after fine-tuning, and it is unclear how well models trained on TextAtlas5M generalize to unseen real-world text scenarios.

---

> ### Author Rebuttal · Authors · 2026-03-31
>
> ## Response to Reviewer fs8n
> We thank Reviewer fs8n for the constructive feedback. We address each point below.
>
> ### **W1. Methodological novelty**
> We thank the reviewer for the comment regarding methodological novelty. However, We would like to clarify that methodological contributions are not limited to model design, but also include problem formulation, data construction, and evaluation protocols, especially for emerging capabilities such as dense and long-form text rendering.
>
> In this work, we
> - identify dense long-text image generation as a distinct and underexplored setting, supported by the large gap in token length between existing datasets (e.g., 9.92 in AnyWords3M and 16.13 in Mario 10M vs. 148.82 in TextAtlas5M),
> - design a structured multi-stage data pipeline (including SAM2-guided region detection and perspective-aware rendering) to generate diverse and controllable training data beyond simple collection,
> - introduce TextAtlasEval and a layout-aware metric to better capture structural consistency beyond OCR-based evaluation.
>
> Importantly, our experiments show that these design choices lead to substantial and consistent improvements across multiple model families and external benchmarks. For example, Pixart-α improves Text Score on LongText-Bench from 0.14 to 24.28, and Word Accuracy on CVTG-2K from 0.02 to 31.30 (Table 4).
>
> We believe that establishing the correct data and evaluation paradigm is a necessary and impactful step toward future algorithmic advances in this area.
>
> ### **W2. Uncover specialized domains**
> We thank the reviewer for this insightful comment. We agree that certain specialized domains remain underexplored, and we have explicitly acknowledged this in our conclusion.
>
> TextAtlas5M already provides partial coverage of scientific content through subsets such as Paper2Text (from arXiv dataset) and PPT2Structured. However, broader multilingual support and formula-intensive layouts are not yet fully covered in the current version of this work, and we consider these important directions for future extension.
>
> As a general-purpose dataset, our goal is to provide a scalable and diverse foundation for long-text image generation rather than exhaustive domain coverage. We hope the community can build upon TextAtlas5M to address these specialized needs.
>
> ### **W3. Model performance on unseen text scenarios**
>
> We thank the reviewer for this important question. We would like to point out that our evaluation already includes several experiments specifically designed to assess generalization to unseen scenarios.
>
> Beyond TextAtlasEval, we evaluate our fine-tuned models on two independently curated text benchmarks, LongText-Bench and CVTG-2K (Table 4). These benchmarks are independently curated and not used during training, and our models show substantial improvements on both, e.g., Pixart-α improves Text Score on LongText-Bench from 0.14 to 24.28 and Word Accuracy on CVTG-2K from 0.02 to 31.30, **demonstrating generalization beyond our own benchmark**.
>
>
> ### **Questions 1. How ensure the quality of text**
> We thank the reviewer for this question. We adopt different processing strategies for different subsets to ensure the quality and correctness of rendered text across the dataset:
> * For Paper2Text and PPT2Structured: We use PyMuPDF to extract text elements along with structural information from the source PDF files. This direct extraction ensures high accuracy of the text content.
> * For CleanTextSynth, TextVisionBlend, and StyledTextSynth: Since the text is programmatically rendered into images, the corresponding OCR annotations are guaranteed to be 100% accurate by construction.
> * For real-world data: We employ a multi-stage processing pipeline. Detailed descriptions of this process are provided in Appendix H.
>
> ### **Questions 2. Model finetuned on other data**
> We thank the reviewer for this suggestion. We note that our paper already includes controlled comparisons that address this concern.
> * In Table 6, using the same model (PixArt-α), we compare fine-tuning on TextAtlas5M vs. Mario10M on LAIONEval4000. TextAtlas5M achieves significantly better performance (Acc: 67.53 vs. 52.37), **demonstrating that improvements do not come from data scale alone**.
>
> * To further support this, we provide a direct comparison of PixArt-α fine-tuned on TextAtlas5M vs. Mario10M on TextAtlasEval. As shown in the table below, TextAtlas5M  outperforms Mario10M, with particularly large gaps on long-text scenarios. This directly demonstrates that the improvements stem from the specific design of TextAtlas5M rather than simply the presence of additional text data.
>
>     | Data | CleanTextSynth Acc | StyleTextSynth Acc |
>     |--------------|-------------------|-----------------|
>     | base| 0.23 | 0.05 |
>     | + Mario 10M | 3.72 | 1.24 |
>     | + TextAtlas5M | **39.14** | **22.10** |
>
>
> These comparisons together suggest that the gains are attributable to the specific properties of TextAtlas5M.

---

> > ### Author Rebuttal · Reviewer_fs8n · 2026-04-02
> >
> > Thanks for the response. My concerns have been addressed.

---

### Official Review · Reviewer_LUp3 · 2026-03-13

**Soundness:** 3
**Presentation:** 3
**Significance:** 2
**Originality:** 3
**Overall Recommendation:** 5
**Confidence:** 3

**Summary:**

This paper introduces TextAtlas5M, a large-scale dataset of 5 million images designed for long-form and structured text image generation. It includes both synthetic and real-world images across multiple domains including slides, posters, infographics, papers, and book covers and is paired with a benchmark, TextAtlasEval, consisting of 4,000 human-refined samples for evaluation. The dataset and benchmark enable fine-tuning of existing open-source and proprietary models, demonstrating notable improvements in OCR accuracy, character-level F1, and layout consistency.

**Compliance With Llm Reviewing Policy:**

Affirmed.

**Final Justification:**

My questions were addressed and I will maintain the score.

**Key Questions For Authors:**

Refer to the Weaknesses

**Limitations:**

yes

**Strengths And Weaknesses:**

Strengths
1. TextAtlas5M is one of the largest datasets targeting long-text image generation, covering multiple domains and both synthetic and real-world images.
2. The dataset enables fine-tuning of models that substantially improve text rendering performance, providing a concrete resource for advancing research in text-conditioned image generation showcased by an example of Pixart-$alpha$ and Lumina-mgpt.
3. TextAtlasEval is human-refined and covers diverse layout and text density scenarios, offering a rigorous evaluation framework.
4. The authors provide transparency about dataset construction, subset distinctions, annotation methods, and filtering procedures.

Weaknesses

Most of the questionable parts are addressed in the main paper as well as the Appendix. Several concerns/questions would be
1. Regarding the four splits which constitutes this dataset, which part contributes to which text generation capability i.e. which split is most effective when used as finetuning data.
2. Despite layout-aware metrics like IOUCount is proposed, can advanced qualitative measures be considered? i.e. the text being generated can lose aesthetic/structural consistency, cohesiveness. fonts style and spacing can be changed and corrupted in many cases of T2I.
3.  The authors claim that multi-linguality is not considered, but they mention the dataset primarily consists of English and Chinese. How does Chinese perform compared to English? Any details in composition and evaluation analysis?
4. The community can also benefit from the author's intuitions/observations on how to improve the text generation capabilities for the T2I models other than simply finetuning on this TextAtlas5M dataset.

Suggestions
1. Main figure 2. is too dense. Maintaining content can be fine but the font size in each bullet points can be increased and the overall readability can be improved.

---

> ### Author Rebuttal · Authors · 2026-03-31
>
> ## Response to Reviewer LUp3
> We sincerely thank Reviewer LUp3 for the positive assessment and constructive feedback. We address each point below.
>
> ### **W1. Which subset contributes to which text generation capability**
>
> We thank the reviewer for this insightful question. While a controlled per-subset ablation is not included in the current version, Table 2 provides indicative evidence by evaluating model performance across different subsets, each corresponding to a distinct data distribution and capability.
>
> From these results, we observe the following trends:
> - CleanTextSynth is associated with strong improvements in basic text rendering accuracy (Lumina-mgpt Acc 0.12 to 48.55), suggesting that clean synthetic text provides effective supervision for foundational text fidelity.
> - TextVisionBlend shows substantial gains on interleaved scenarios (0.00 to 65.34), indicating improved capability in handling structured text-image layouts.
> - StyledTextSynth is linked to scene-text rendering, where performance is more sensitive to resolution and visual integration.
> - TextScenesHQ remains the most challenging subset, reflecting real-world complexity in both layout and semantics.
>
> Additionally, Table 7 provides further evidence: mixed training (synthetic + real) consistently outperforms real-only training, indicating that **synthetic and real subsets provide complementary supervision signals**.
>
> Together, these observations suggest a progressive learning pattern, where simpler synthetic data supports foundational capabilities, and more complex subsets contribute to layout understanding and real-world generalization.
>
> ### **W2. advanced qualitative metrics**
> We thank the reviewer for highlighting the importance of aesthetic and structural consistency in text rendering. To better assess these aspects, we use **GPT-4o as an MLLM evaluator** to score Text Aesthetics (TA) and Text Consistency (TC) on a scale of 1–5, following a similar protocol to T2I-CoReBench. Results on the TextVisionBlend subset are shown below:
>
> |  | TA | TC |
> |-|-|-|
> | SD-3.5| 1.24 | 0.73 |
> | PixArt-α | 0.01 | 0.04 |
> | PixArt-α(finetuned) | 3.68 | 4.03 |
> | Nano Banana | 4.17 | 4.52 |
>
> Models fine-tuned on TextAtlas5M achieve substantial improvements in both aesthetic and structural dimensions, suggesting that our dataset improves not only textual accuracy but also visual quality. We will include this evaluation protocol and expanded results in the revised version.
>
> ### **W3. Multi-linguality**
> We thank the reviewer for this question. Our dataset contains Chinese images primarily in the AnyText-derived (LongWordsSubset-A) subsets, accounting for approximately 3% of the total samples. Due to the relatively small proportion of Chinese samples, we did not conduct a separate Chinese-specific evaluation. We acknowledge this as a limitation and will include language-stratified analysis in the revised version.
>
> ### **W4. Observations on how to improve the text generation capabilities.**
>
> We thank the reviewer for inviting broader discussion, and are happy to share some observations and intuitions based on our experimental findings.
>
> We find that, under simplified settings **without complex backgrounds or layouts, rendering textual content is relatively easy for current T2I models to learn** — after training on corresponding data, models can produce reasonably accurate text renderings.
>
> The challenging part is mixed text-and-natural-image generation (e.g., posters, infographics, product images), where text quality tends to degrade significantly.
>
>
> We hypothesize that a key underlying reason is the relative sparsity of text supervision: in mixed-content images, text regions occupy only a small fraction of pixels, **leading the model to prioritize overall visual appearance aligned with the prompt, rather than precise text rendering**.
>
> Based on this observation, we believe that improving text generation requires not only scaling data but also strengthening supervision on text regions. For example, one possible method is to upweight the loss on text regions to strengthen the supervision.
>
> Beyond supervision, we also observe that **image tokenizers used in current T2I models may represent another critical bottleneck**. As shown in Section E1 of our supplementary material (Figures 13 and 14), image tokenizers exhibit reconstruction errors on text-containing images. The tokenizer capacity for representing fine-grained text details remains insufficient.
>
> Taken together, we believe that improving text rendering quality requires systematic advances on two fronts: supervision mechanisms and representational capacity (e.g., tokenizers). We will include this discussion in the revised paper as a concrete outline of future research directions.
>
> ### **S1. Fig2 refinement**
> We thank for the helpful suggestion. We will refine the Fig.2 in the final version.

---

> > ### Author Rebuttal · Reviewer_LUp3 · 2026-04-02
> >
> > I keep the score as is.

---

### Decision · Program_Chairs · 2026-04-30

**Decision:**

Accept (regular)

**Comment:**

This paper presents TextAtlas5M, a large scale dataset for text to image generation focused on images that contain long, dense, and structured text, together with TextAtlasEval, a human refined benchmark of 4000 examples. The motivation is that existing datasets and benchmarks emphasize short and simple text, while many practical settings such as slides, posters, documents, and interleaved layouts require richer text rendering and stronger layout control. To address this gap, the authors assemble a diverse mixture of synthetic and real world sources and unify them into a training dataset intended for dense text image generation. The paper also evaluates a range of open and closed source models and shows that current systems still struggle on this setting, especially on complex layouts and realistic scenes. Fine tuning on TextAtlas5M yields gains for both autoregressive and diffusion style models on OCR based metrics, layout related measures, and external benchmarks.


The reviewers agree that this is an important and timely problem, and they view the scale, diversity, and benchmark construction as strengths of the submission. They also note that the empirical improvements are clear and that the paper offers a useful resource for the community, even if the main contribution is primarily in dataset and evaluation design rather than in new modeling methods. The main concerns were on missing related work, limited analysis of specialized domains, annotation quality and subset specific contributions, and the fact that the strongest demonstrated gains are on models with relatively weak text rendering ability. The rebuttal addresses a meaningful portion of these concerns by clarifying expected roles of different subsets, pointing to external benchmark improvements as evidence of generalization, and offering additional discussion points. Overall, I am inclined to recommend acceptance because the paper identifies a real gap, provides a substantial new dataset and benchmark, and demonstrates practical impact on an under-explored but important capability.